# Retrieval of NO$_2$ profiles from three years of Pandora MAX-DOAS measurements in Toronto, Canada

Ramina Alwarda[1,2], Kristof Bognar[3], Xiaoyi Zhao[2], Vitali Fioletov[2], Jonathan Davies[2], Sum Chi Lee[2],

Debora Griffin[2], Alexandru Lupu[2], Udo Frieß[4], Alexander Cede[5,6], Yushan Su[7], Kimberly Strong[1]

[1]Department of Physics, University of Toronto, Toronto, Canada
[2]Air Quality Research Division, Environment and Climate Change Canada, Toronto, Canada
[3]3vGeomatics, Vancouver, Canada
[4]Institute of Environmental Physics, Heidelberg University, Heidelberg, Germany
[5]NASA Goddard Space Flight Center, Greenbelt, USA
[6]LuftBlick, Innsbruck, Austria
[7]Ontario Ministry of the Environment, Conservation and Parks, Toronto, Canada

*Correspondence to*: Ramina Alwarda (ramina.alwarda@mail.utoronto.ca)

**Abstract.** The purpose of this work is to derive new NO$_2$ vertical profiling data products from Pandora spectrometers and investigate the factors contributing to the bias of this dataset relative to established ground-based and spaceborne datasets. Possible applications of the NO$_2$ vertical profile dataset include air quality monitoring and satellite validation studies. We explore the application of the optimal estimation method to Pandora multi-axis differential optical absorption spectroscopy (MAX-DOAS) measurements to retrieve vertical profile information for nitrogen dioxide (NO$_2$). We use the Heidelberg Profile (HeiPro) retrieval algorithm to derive, for the first time, NO$_2$ profiles and partial columns (0–4 km) from Pandora MAX-DOAS measurements from 2018–2020 at Downsview, a suburban neighbourhood in the north end of Toronto, Canada that is subject to local traffic emissions and urban influences. Validation of the new dataset was done via comparison with official Pandora direct-Sun measurements, in situ observations, satellite data, and an air quality forecasting model. We find that, for tropospheric partial column comparisons, the HeiPro dataset has a positive mean relative bias to Pandora direct-Sun (61% $\pm$ 9.7%) and TROPOMI (37% $\pm$ 51%) observations, as well as the GEM-MACH model output (67% $\pm$ 7.1%), with similar seasonal and diurnal cycles in the bias with Pandora direct-Sun and GEM-MACH. Contributing factors to the large bias of HeiPro to Pandora direct-Sun were investigated, and NO$_2$ heterogeneity, combined with differences between direct-Sun and multi-axis viewing geometries, was found to contribute a maximum of 52% of the total relative bias during morning measurement times. For surface NO$_2$ comparisons, we find that HeiPro measurements capture the magnitude and diurnal variability of surface NO$_2$ reasonably well (mean relative bias to in situ surface NO$_2$: $-9.7\% \pm 7.5\%$) but are low-biased compared to GEM-MACH (mean relative bias: $-37\% \pm 2.4\%$). Compared to HeiPro, the GEM-MACH model profiles are high-biased in the lower boundary layer and low-biased in the free troposphere.

## 1 Introduction

Nitric oxide (NO) and nitrogen dioxide ($NO_2$), together comprising $NO_x$, play important roles in air quality, tropospheric chemistry, and stratospheric ozone ($O_3$) chemistry. In the troposphere, $NO_x$ is primarily emitted from fossil fuel combustion and nitrogen fertilizers and has a near-surface peak in the profile shape in polluted regions; additionally, it is emitted from natural sources such as lightning and biomass burning (e.g., ECCC 2016; Murray, 2016). The photolysis of $NO_2$ rapidly contributes to the production of $O_3$ and therefore photochemical smog formation. $NO_2$ also plays a more direct role in air quality as it has toxic biological effects at high concentrations and has been linked to negative health outcomes such as asthma exacerbations (ECCC, 2016). Tropospheric $NO_2$ has a large spatiotemporal variability and this can be attributed to its short lifetime, variability of emissions, and meteorological variations (Beirle et al., 2003).

Due to its role in various atmospheric processes and impact on air quality, it is important to continuously monitor $NO_2$ and its spatiotemporal variability. Surface $NO_2$ is measured using in situ photolytic conversion/chemiluminescence as part of Canada's Air Quality Health Index (AQHI, Stieb et al., 2008) along with surface $O_3$ and surface fine particulate matter with diameter $< 2.5$ microns (PM2.5). While in situ instruments provide surface measurements with high temporal resolution, such measurements are spatially limited both in geographic extent and altitude. Ground-based remote sensing measurements address the latter limitation by measuring $NO_2$ throughout the troposphere and stratosphere and are used in air quality and satellite validation studies (e.g., Ma et al., 2013; Griffin et al., 2019; Zhao et al., 2019, 2020; Verhoelst et al., 2021). Non-geostationary satellite instruments, on the other hand, address spatial limitations from ground-based monitoring and provide global coverage. A suite of UV-visible satellite instruments has monitored $NO_2$ columns since 1995, beginning with the Global Ozone Monitoring Experiment (GOME, Burrows et al., 1999) and more recently, the TROPOspheric Monitoring Instrument (TROPOMI) onboard the European Space Agency's Sentinel-5 Precursor (S5P) satellite. While such satellite instruments provide global coverage, the temporal frequency of at any location is limited, which hinders insight into the spatiotemporal variation of $NO_2$ on large scales. The recent additions of geostationary satellite instruments such as the Tropospheric Emissions: Monitoring of Pollution (TEMPO, Zoogman et al., 2017) and the Geostationary Environmental Monitoring Spectrometer (GEMS, Kim et al., 2020) address this limitation of low-Earth satellite orbits. Thus, in situ, ground-based, and spaceborne measurements all contribute to our understanding of the spatiotemporal variability of $NO_2$.

Ground-based UV-visible spectra are usually analysed using Differential Optical Absorption Spectroscopy (DOAS, Platt and Stutz, 2008). These measurements include zenith-scattered-light DOAS (ZSL-DOAS), which is most sensitive to the stratospheric column, multi-axis DOAS (MAX-DOAS), which is most sensitive to the tropospheric column, and direct-Sun (DS), which measures the total column with equal sensitivity to the troposphere and stratosphere. One such ground-based instrument that performs routine direct-Sun, zenith-sky, and multi-axis measurements of the atmosphere is the Pandora UV-visible spectrometer (Herman et al., 2009). Pandora instruments are distributed globally as part of the Pandonia Global

Network (PGN), an international collaboration led by the National Aeronautics and Space Administration (NASA), European Space Agency (ESA), the US Environmental Protection Agency (US-EPA), and the Korean National Institute of Environmental Research (NIER), which provides standardized high-quality datasets, such as direct-Sun $NO_2$ total columns, for various purposes including satellite validation and air quality assessments (e.g., Zhao et al., 2022).

Additionally, Pandora MAX-DOAS spectra can be used to retrieve vertical profiles of trace gases in the lower troposphere, for example by using the optimal estimation method (OEM), a maximum a posteriori profile inversion algorithm which is based on Bayesian statistics and provides a robust error analysis (Rodgers, 2000; Frieß et al., 2019). Utilizing a MAX-DOAS $NO_2$ dataset is advantageous because it provides sensitivity to tropospheric absorbers, profile information, increased temporal resolution compared to spaceborne measurements, horizontal distribution of $NO_2$ surrounding a measurement site 75 (Dimitropoulou et al., 2020), and a horizontal resolution that is more suited for assessing model performance compared to in situ data (Blechschmidt et al., 2020). $NO_2$ profiles from MAX-DOAS measurements are also a useful tool for aiding in satellite retrieval algorithms and validation; previous studies have utilized $NO_2$ profiles derived from MAX-DOAS data as inputs for the TROPOMI retrieval algorithm to reduce the systematic underestimation in the satellite data compared to ground-based data (e.g., Dimitropoulou et al., 2020). Partial columns of $NO_2$ from MAX-DOAS data have also been used in ground-based 80 validation of spaceborne measurements, whereby satellite tropospheric $NO_2$ is typically biased low to ground-based MAX-DOAS $NO_2$ in polluted regions (e.g., Pinardi et al., 2020; Verhoelst et al., 2021). Satellite measurements rely on ground-based data to continuously improve satellite algorithms and understand the bias that exists between spaceborne and ground-based measurements as a function of location, especially due to the difficulty in implementing location-specific a priori $NO_2$ profiles in satellite retrieval algorithms (Verhoelst et al., 2021). In addition to OEM, there are various other retrieval techniques that 85 convert MAX-DOAS measurements to $NO_2$ profiles and subsequent data products. For example, Pandora MAX-DOAS spectra can be used to retrieve $NO_2$ vertical profiles, tropospheric columns, and surface values using an L2 Air-Ratio Sky Algorithm developed by Elena Spinei and the PGN (see PGN software manual v1.8 available at https://www.pandonia-global-network.org/home/documents/manuals/ for further information, last access 13 September 2024). Comparisons of these PGN sky algorithm data products with other datasets at the measurement site in this study will be the subject of a future study.

While OEM has been applied to Pandora MAX-DOAS data in a study by Nowlan et al. (2018) that compared aircraft retrievals of $NO_2$ to ground-based MAX-DOAS and direct-Sun Pandora measurements for an eleven-day period, it has not yet been applied to Pandora MAX-DOAS observations for extended periods of time. A longer intercomparison period would enable investigations of annual and seasonal trends compared to other $NO_2$ datasets, especially comparing Pandora MAX-95 DOAS to direct-Sun observations. Previous comparisons between $NO_2$ observations of MAX-DOAS to direct-Sun are limited but would be useful in addressing the difference in the relative biases of multi-axis and direct-Sun observations to satellite measurements. For example, Pinardi et al. (2020) found a bias of 10–15% in the ground-based MAX-DOAS $NO_2$ columns compared to direct-Sun tropospheric columns from the same station, but such comparisons were during the overpass times of

the two satellites in the study. The relative bias was attributed to NO$_2$ heterogeneity as well as the differences in airmasses
measured by the direct-Sun and multi-axis geometries. Performing NO$_2$ comparisons of MAX-DOAS to direct-Sun throughout
the day would be an important consideration for the validation of geostationary satellite NO$_2$ measurements, since both are
used for the validation of spaceborne tropospheric NO$_2$ (e.g., Verhoelst et al., 2021).

Therefore, due to the advantages of obtaining NO$_2$ profiles, as well as the limited studies comparing MAX-DOAS to
direct-Sun NO$_2$, the aim of this work is to derive NO$_2$ profiles from Pandora MAX-DOAS measurements using OEM and to
assess the quality of the dataset by comparison to established datasets. We present, for the first time, a three-year dataset of 0–
4 km NO$_2$ profiles from 2018–2020 at Downsview, a Pandora measurement site in the north part of Toronto, Ontario, Canada,
located in a suburban area subject to local traffic emissions and urban influences. To our knowledge, this work is the first study
to apply the optimal estimation method to Pandora MAX-DOAS data to retrieve a multi-year NO$_2$ profile dataset. We assess
the biases and contributing factors to the biases between the OEM-based Pandora dataset versus (1) standard measurements
comprising Pandora direct-Sun (Pandora-DS) and TROPOMI tropospheric columns as well as in situ surface NO$_2$, and (2)
profiles, partial columns, and surface NO$_2$ from the Global Environmental Multiscale–Modelling Air-quality and Chemistry
(GEM-MACH) high-resolution regional air quality forecast model. Where possible, we quantify the contributing factors to the
biases and assess the impact of spatial heterogeneity on the biases. The resulting OEM-based Pandora NO$_2$ profile dataset can
used to assess spatiotemporal NO$_2$ heterogeneity around the measurement site as well as any biases between spaceborne and
ground-based NO$_2$ at the measurement site.

The paper is organized as follows: Section 2 describes the various instruments, algorithms, and models used in this
study, as well as the data filtering and co-location criteria that were applied. In Section 3, the quality of the OEM-based Pandora
dataset is assessed by comparing it with Pandora-DS and TROPOMI tropospheric columns as well as GEM-MACH partial
columns. Section 4 presents the surface NO$_2$ volume mixing ratio (VMR) comparisons among the OEM-based Pandora dataset,
in situ data from the National Air Pollution Surveillance (NAPS) monitoring program, and GEM-MACH data, as well as the
comparisons between Pandora OEM and GEM-MACH NO$_2$ profiles. Seasonal and diurnal trends are examined where possible,
and the causes of differences among the measurements are investigated. Finally, Section 5 summarizes the implications and
conclusions.

## 2 Methods: Instruments, datasets, and models

An overview of the instruments and corresponding datasets used in this study is presented in Fig. 1. Section 2.1
describes the Pandora spectrometer and Sections 2.1.1–2.1.3 describe the direct-Sun, MAX-DOAS, and profiling retrieval
algorithms, respectively. Sections 2.2 and 2.3 describe the TROPOMI and photolytic conversion/chemiluminescence in situ
measurements, respectively. Section 2.4 describes the models used in this work, both auxiliary models used in producing other

datasets (2.4.1, Pratmo and 2.4.2, ERA5) and the model that contributed an additional NO₂ comparison (2.4.3, GEM-MACH). Table 1 provides a summary of the datasets used in the study as well as some characteristics of the datasets such as resolutions and uncertainties.

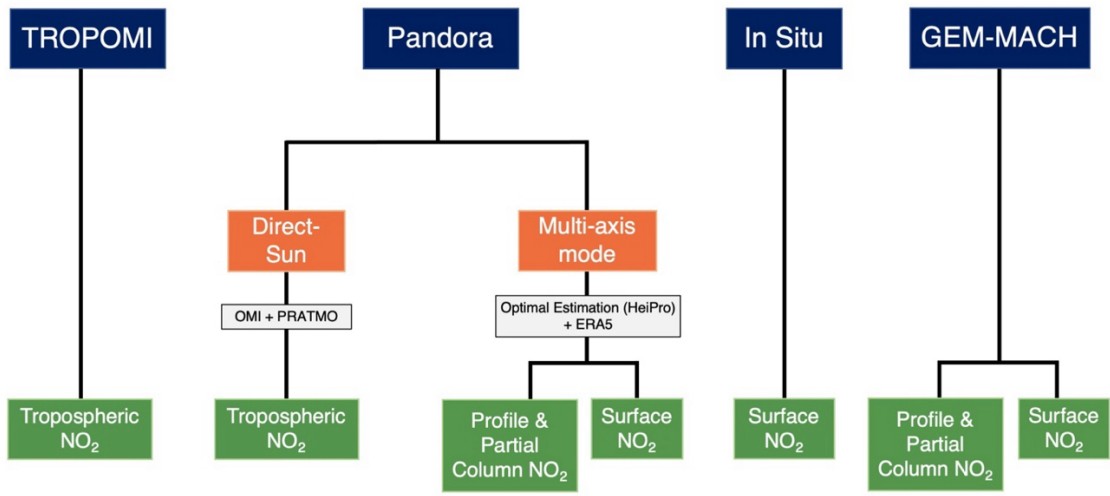

**Figure 1: Overview of the NO₂ data products used in this study. The blue boxes indicate the instrument or model, the orange boxes indicate Pandora viewing geometries, and the green boxes indicate the final data products.**

**Table 1. Overview of the key attributes of the NO₂ datasets used in this study.**

| Dataset | Temporal resolution | Horizontal resolution | Errors/uncertainties |
|---|---|---|---|
| **Pandora-DS total column NO₂** | 90 s | < 4 km (for SZA < 50°) 4–17 km (for SZA 50°–80°) (Herman et al., 2009) | $1.3\times10^{15}$ molec/cm² (Herman et al., 2009, 2018) |
| **OMI stratospheric NO₂** | 1 day | 13 km × 24 km (Krotkov et al., 2017) | $2\times10^{14}$ molec/cm² (Krotkov et al., 2017) |
| **HeiPro** | 22 minutes | 5–10 km (effective pathlength, Ortega et al., 2015) | $4.4\times10^{14}$ molec/cm² (this work) |
| **TROPOMI tropospheric NO₂** | 1 day | 7 km × 3.5 km (5.5 km × 3.5 km since Aug. 2019) | $8.5\times10^{14}$ molec/cm² (Eskes and Eichmann, 2019) |
| **GEM-MACH** | 1 hour | 10 km × 10 km | N/A |
| **In situ NO₂** | 1 minute | Point measurement | 0.4 ppbv (Thermo Scientific, 2015) |

## 2.1 Pandora UV-visible spectrometer

Pandora instruments utilize a temperature-stabilized Czerny-Turner spectrometer with a grating of 1200 gr/mm and a back-thinned charge-coupled device (CCD) detector to record spectra between 280–530 nm with a spectral resolution of 0.6 nm. All Pandora $NO_2$ data products presented in this work were obtained from Pandora no. 103 (P103) located on the rooftop of the Environment and Climate Change Canada (ECCC) five-storey building at Downsview (43.78 °N, 79.47 °W, 186 m a.s.l.) in Toronto, Canada. This building is located in a suburban region with several nearby roads. P103 operates in direct-Sun, zenith-sky, and multi-axis viewing geometries during the sunlit period. Figure 2 shows the May 2018 – December 2020 TROPOMI tropospheric $NO_2$ field at the measurement site and surrounding region, as well as the differences in viewing directions between the direct-Sun and multi-axis measurements. Direct-Sun and multi-axis data products from P103 from February 2018 to June 2020 were used in this study. Further details on the retrieval algorithms used on the Pandora data are discussed in Sections 2.1.1 (direct-Sun), 2.1.2 (multi-axis), and 2.1.3 (profiling).

### 2.1.1 Pandora-DS retrievals

The standard Pandora-DS total column $NO_2$ data product is obtained using Total Optical Absorption Spectroscopy (TOAS), as implemented by PGN's BlickP software (Cede, 2019). Direct-Sun spectra in the 400–440 nm range are fitted with cross-sections of $NO_2$ (at an effective temperature of 254.5 K, Vandaele et al., 1998), $O_3$ (at an effective temperature of 255 K, Brion et al., 1993, 1998; Daumont et al., 1992), and a fourth-order polynomial to produce SCDs of $NO_2$ with a clear-sky precision of $2.7 \times 10^{14}$ molec $cm^{-2}$ (Herman et al., 2009). A synthetic reference spectrum is used in the analysis and is obtained by taking an average of several measured spectra which are corrected for their total optical depth. Following this, $NO_2$ SCDs are converted to vertical column densities (VCDs or total columns) using geometric AMFs. The Pandora-DS $NO_2$ VCD has an absolute accuracy of $1.3 \times 10^{15}$ molec $cm^{-2}$ (Herman et al., 2009). The Pandora-DS $NO_2$ VCDs used in this study are obtained from the PGN (https://data.pandonia-global-network.org/Downsview/Pandora103s1/, file version: rnvs1p1-7, last access: 16 April 2024) and only high-quality data (i.e., L2 flags of 0 and 10) were used. Because the focus of this work is on tropospheric $NO_2$ intercomparisons, the resulting $NO_2$ VCDs were converted to tropospheric $NO_2$ columns by subtracting the stratospheric $NO_2$ columns obtained from the Ozone Monitoring Instrument (OMI, NASA standard product version 3.0 level 2, Krotkov et al., 2017) onboard the Aura satellite. Due to the diurnal variation of $NO_2$, and the satellite's overpass time of 13:30 local time (LT), a photochemical box model (Pratmo, discussed further in Section 2.3.1) was used to calculate stratospheric $NO_2$ at various Pandora measurement times throughout the day. The stratospheric portion that was removed accounted for 34% $\pm$ 2.8% of the Pandora-DS $NO_2$ total columns. The Pandora-DS tropospheric $NO_2$ data were averaged to obtain hourly means for comparison with the other datasets.

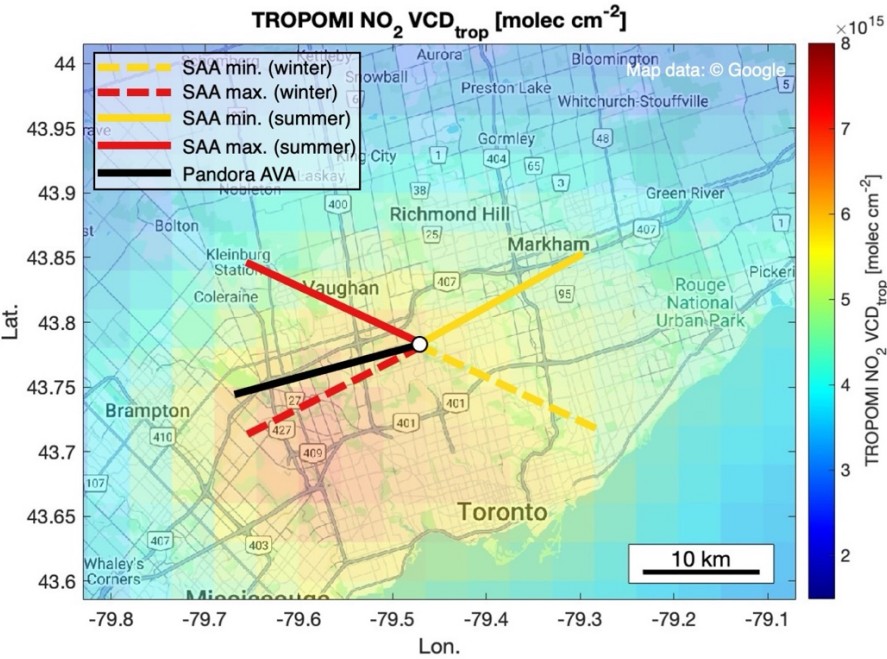

**Figure 2: A map (© Google Maps) of the measurement site (Downsview, white circle) and surrounding Greater Toronto Area masked by the TROPOMI pixel-averaged tropospheric NO₂ field from May 2018 – December 2020. The color bar indicates the NO₂ tropospheric column in molec cm⁻². Solid and dashed lines represent the minimum (yellow) and maximum (red) solar azimuth angle (SAA) during summer and winter, respectively. The solid black line represents the Pandora instrument's multi-axis azimuth viewing angle (AVA) of 255°.**

### 2.1.2 Differential slant column density (dSCD) retrievals

The multi-axis viewing geometry can provide vertical information from tropospheric absorbers by measuring scattered sunlight at various elevation viewing angles (EVAs) of the instrument. P103 performs both 'long' and 'short' multi-axis scans in an alternating manner, with an azimuth viewing angle of 255° (see Fig. 2). Long scans consist of EVAs of 1°, 2°, 3°, 5°, 8°, 10°, 15°, 20°, 30°, 40°, 50°, and 90° and short scans consist of EVAs of 1°, 2°, 15°, 30°, and 90°. The resulting multi-axis spectra from P103 (i.e., corrected Level 1 data, data file version: smca1c1p1-7) were obtained from the PGN. Utilizing the DOAS technique and fitting parameters outlined in Appendix A of Kreher et al. (2020), spectral fitting to the ratio of two spectra, with one spectrum at each EVA and the other at a 90° reference EVA, is performed to retrieve the differential slant column density (dSCD) of the oxygen collision complex (O₄) and NO₂ at each EVA. This is achieved using the QDOAS software (https://uv-vis.aeronomie.be/software/QDOAS/, last access: 16 April 2024), developed by the Royal Belgian Institute for Space Aeronomy (Danckaert et al., 2017). O₄ and NO₂ dSCDs were retrieved in both the ultraviolet (UV, 338–370 nm, only measurements with the UV band pass filters) and visible (vis, 425–490 nm) windows. Differential cross-sections of NO₂

at 294 and 220 K for both windows (Vandaele et al., 1998), $O_4$ at 293 K for both windows (Thalman and Volkamer, 2013), $O_3$ at 223 and 243 K for UV and 223 K for vis (Serdyuchenko et al., 2014), BrO at 223 K for UV only (Fleischmann et al., 2004), HCHO at 297 K for UV only (Meller and Moortgat, 2000), and $H_2O$ for vis only (Rothman et al., 2010) were convolved using the instrument slit function and the nominal wavelength calibration file from PGN. A fifth-degree polynomial, linear offset, and first-order shift and stretch were used in both windows (Kreher et al., 2020). Following the QDOAS analysis, $O_4$ and $NO_2$ dSCDs with fitting residual root-mean-square (RMS) values $\geq 3\times10^{-3}$ were discarded (only 5% of the data, mostly at SZA > 80°). This filter was selected to improve the quality of the fits and discard noisier fits from the analysis. Next, we discuss the Heidelberg Profile Retrieval algorithm (HeiPro, Frieß et al., 2011; Yilmaz, 2012), which was used to produce aerosol extinction and $NO_2$ profiles.

### 2.1.3 HeiPro retrievals

HeiPro was used to convert Pandora $O_4$ and $NO_2$ dSCDs of at various EVAs to aerosol extinction and $NO_2$ profiles, respectively, from 0–4 km. HeiPro is a maximum a posteriori optimal estimation method (Rodgers, 2000) that utilizes a forward model (a radiative transfer model or RTM, here being SCIATRAN version 2.2, Rozanov et al., 2005), to simulate dSCDs based on a priori $NO_2$ profile information and additional parameters such as aerosol extinction, aerosol optical properties, surface albedo, and temperature. Exponentially decreasing aerosol extinction and $NO_2$ profiles with scale heights of 1 km and column values of 0.18 (aerosol optical depth, AOD) and $9\times10^{15}$ molec $cm^{-2}$ ($NO_2$) were used as a priori information in the HeiPro simulation. Additional a priori inputs to HeiPro were daily pressure and temperature profiles from ERA5 reanalysis data at the grid box nearest to the measurement site (discussed further in Section 2.4.2). For the aerosol retrievals, a single scattering albedo value of 0.92 and asymmetry factor of 0.68 were used. For all retrievals, a surface albedo value of 0.06 was used, and correlation matrices with a variance of 50% and correlation length of 200 m were used.

Given these settings, a cost function is used to assess the difference between the dSCDs simulated by the RTM and the measured dSCDs while accounting for additional a priori constraints. The Levenberg–Marquardt algorithm is then used to iteratively find the minimum of the cost function. At this minimum, the most probable atmospheric state or the maximum *a posteriori* estimate (MAP) is then taken to be the $NO_2$ profile. HeiPro utilizes a two-step approach: prior to each $NO_2$ profile retrieval, aerosol extinction profiles are retrieved based on $O_4$ dSCDs; these are subsequently used as inputs for the $NO_2$ profile retrievals. Profiles were retrieved separately for UV and visible scans, with RTM wavelengths of 360 nm for $O_4$-UV and $NO_2$-UV, 477 nm for $O_4$-vis, and 460 nm for $NO_2$-vis. Full details on the HeiPro algorithm can be found in Frieß et al. (2006, 2011). The profile grid is from 0–4 km, with a vertical resolution of 200 m. Therefore, the $NO_2$ concentration for the 0–200 m grid level represents the average throughout this range, i.e., the concentration at 100 m. The HeiPro 'surface' concentration reported in this study is then obtained by extrapolating the VMR at the 0–200 m grid level to 0 m for each individual profile.

The linear extrapolation method used in this study produces $NO_2$ surface values that are, on average, 1 ppbv larger than the $NO_2$ value at 100 m. Therefore, this extrapolation method may underestimate the HeiPro surface $NO_2$ values that we report.

Additionally, only the long multi-axis scans (see Section 2.1.2 for list of EVAs) recorded in the UV window, together termed the 'long UV' scans, were incorporated in this work as they have the most reliable results. We found the HeiPro long UV results to be better suited for this study for the following reasons: (1) as expected, the degrees of freedom for signal (DOFS) for each profile, which represents the number of independent pieces of information obtained from the measurements, are greater for the long scans (2.35 and 3.13 for UV and vis, respectively) compared to the short scans (2.03 and 2.75 for UV and vis, respectively), see Fig. A1 (in Appendix A). (2) The $NO_2$ partial column errors are smaller for the long scans ($5.45 \times 10^{14}$ and $2.99 \times 10^{14}$ molec/cm$^2$ for UV and vis) compared to the short scans ($7.51 \times 10^{14}$ and $3.49 \times 10^{14}$ molec/cm$^2$ for UV and vis, respectively), see Fig. A1. (3) Lastly, the long UV scans show improved regression slopes, intercept, and correlation coefficient to the NAPS in situ surface VMRs compared to the long vis scans (see Fig. A2). This is likely due to the fact that scans in the visible window cover a greater horizontal region than in the UV (Ortega et al., 2015), and so UV scans may better capture the $NO_2$ field localized to the in situ instrument inlet. Therefore, only HeiPro retrievals with long UV scans have been used in this study for the comparisons with Pandora-DS and TROPOMI tropospheric columns, GEM-MACH partial columns and surface $NO_2$, and NAPS in situ surface $NO_2$. Example MAX-DOAS dSCDs of $O_4$ and $NO_2$, along with averaging kernels for aerosol extinction and $NO_2$ profile retrievals, are demonstrated in Fig. A3.

Lastly, retrievals for which both the $NO_2$ profiles and aerosol extinction profiles had DOFS < 1 were excluded from the analysis for quality control purposes (e.g., Vlemmix et al., 2015) and represented 19% of the dataset. Such retrievals contain limited information from the measurements and are more influenced by the a priori profile. The mean DOFS values before and after filtering are $1.61 \pm 0.68$ and $1.88 \pm 0.42$ for aerosol extinction, respectively, and $2.35 \pm 0.49$ and $2.39 \pm 0.40$ for $NO_2$, respectively. In addition to providing $NO_2$ profiles from 0–4 km, HeiPro also provides an integrated value from 0–4 km, thereby providing a partial $NO_2$ column for the lower troposphere. All HeiPro data products presented here are hourly averages. From here on, surface $NO_2$ as well as 0–4 km partial columns and profiles retrieved from Pandora MAX-DOAS measurements using HeiPro will be described as HeiPro data products.

## 2.2 TROPOMI

The TROPOMI instrument measures nadir solar UV-visible radiance in a sun-synchronous orbit at an altitude of 817 km and a local overpass time of 13:30 LST. The TROPOMI instrument provides spaceborne $NO_2$ measurements at a horizontal resolution of 3.5 km × 5.5 km (3.5 km × 7.5 km prior to August 2019). The DOAS technique is used to convert UV-visible spectra to $NO_2$ slant column densities (SCDs). Following this, a chemical transport model (TM5-MP) assimilates the SCDs to $NO_2$ vertical profiles, from which a stratospheric SCD is determined. Finally, air mass factors (AMFs, which are based on

surface albedo, NO2 profile shape, etc.) are used to obtain a tropospheric column. Details about the instrument and NO2 retrieval scheme can be found in van Geffen et al. (2020). The TROPOMI tropospheric columns used in this work are from processor version 2.3.1 (van Geffen et al., 2022). We used tropospheric NO2 data product quality assurance values (qa_value) ≥ 0.75 to exclude retrievals with very cloudy scenes. Only TROPOMI pixels that were within 10 km of the Pandora

measurement site, i.e., the ECCC building at Downsview, and within ±10 minutes of the Pandora MAX-DOAS measurement time (Zhao et al., 2020) were included in the comparisons.

### 2.3 In situ measurements

The in situ instrument (Thermo 42i) at the Toronto North site, i.e., the Pandora measurement site, monitors surface NO2 under the NAPS Program and is located near a major road. Horizontally, it is located 100 m away from the Pandora

instrument; vertically, it has an air intake inlet that is 4 m above ground level and 11 m below the Pandora instrument. It utilizes the photolytic conversion/chemiluminescence technique to measure NO2 with a precision of 0.4 ppb. The NO2 data collected at the Toronto North site was provided on an hourly-averaged timescale and had undergone final validation by the Ontario Ministry of the Environment, Conservation and Parks.

### 2.4 Models

#### 2.4.1 Pratmo

Pratmo is a photochemical box model with detailed stratospheric chemistry that is of relevance to NO2 (McLinden et al., 2000; Brohede et al., 2008; Lindenmaier et al., 2011). Because the Pandora-DS observations provide NO2 total columns and we are interested in only the tropospheric portion, the stratospheric column must be removed. However, sun-synchronous satellite observations of stratospheric NO2 cannot solely be used to do this because they provide a single observation per day

whereas stratospheric NO2 exhibits diurnal variability and Pandora measures throughout the day. Therefore, Pratmo provides a modelled ratio of stratospheric NO2 at the Pandora measurement time and the OMI overpass time. The measured OMI stratospheric columns can then be multiplied by this modelled ratio to obtain stratospheric NO2 columns at Pandora measurement times. The OMI stratospheric NO2 measurements used in this study are from version 3.0 level 2 (SPv3.0, Krotkov et al., 2017) of NASA's standard product (SP).

#### 2.4.2 ERA5

Various data products from the European Centre for Medium-Range Weather Forecasts (ECMWF) Reanalysis v5 (ERA5) were utilized in this work (Hersbach et al., 2023). As previously mentioned, pressure and temperature profiles were used as a priori information in the HeiPro algorithm that retrieves NO2 profiles from Pandora MAX-DOAS spectra. ERA5 pressure and temperature profiles from 0.1–30 km at the grid box nearest to the Downsview site were averaged at timestamps

of 11:00, 14:00, 17:00, 20:00, and 23:00 UTC to produce daily a priori temperature and pressure profiles. Secondly, hourly

ERA5 2 m temperature and surface pressure data at the grid box nearest to the Downsview site were used to calculate a 15 m NO₂ column (see Section 3 and Equation 1) to account for the rooftop location of the Pandora instrument. Additionally, planetary boundary layer (PBL) height was obtained from ERA5 at the Downsview site to investigate how the bias among datasets varies as a function of PBL height.

**2.4.3 GEM-MACH**

GEM-MACH is ECCC's operational regional air quality model, which provides predictions of pollutants over North America at a 10 km × 10 km horizontal resolution on an hourly timescale for a 72 h period, and is run twice per day at 0 and 12 UTC (Moran et al., 2009; Pavlovic et al., 2016; Pendlebury et al., 2018). The model provides output for 80 vertical levels from the surface to approximately 0.1 hPa. In this work, NO₂ profiles and partial columns from ~0–5.0 km within the model grid box nearest to the Pandora measurement site were extracted from 2018 to 2020, with a 200 m vertical grid. GEM-MACH utilizes emissions generated from inventories from the United States Environmental Protection Agency Air Emissions Modeling Platform, from Canada's Air Pollutant Emission Inventory, and from Mexico's National Emissions Inventory. The model includes physical and chemical processes such as tropospheric gas- and aqueous-phase chemistry, inorganic heterogeneous chemistry, and wet and dry deposition. It should be noted that the operational version of GEM-MACH does not currently include $NO_x$ sources in the free troposphere, such as lightning and aircraft emissions.

**3 HeiPro partial column comparisons**

In this section, we compare the HeiPro partial columns of NO₂ to (i) Pandora-DS tropospheric columns, (ii) TROPOMI tropospheric columns, and (iii) GEM-MACH partial columns. It is important to note that the comparisons are not entirely equivalent because HeiPro provides an integrated partial column from 0–4 km while TROPOMI and Pandora-DS provide tropospheric columns and GEM-MACH provides 0–5 km partial columns. Figure 3 displays the following scatter plots: HeiPro partial columns versus (a) Pandora-DS tropospheric columns, (b) TROPOMI tropospheric columns, and (c) GEM-MACH partial columns. Three types of linear regressions are presented in Fig. 3: the first is the York linear fit (York et al., 2004) in which the uncertainties in both datasets are incorporated into the regression, the second is the zero-intercept linear regression, and the third is the ordinary least squares linear regression (used only for the HeiPro vs. GEM-MACH regression since error estimates were not available for GEM-MACH data). The multiplicative bias is obtained from the zero-intercept slope, and is presented as a deviation, in %, from a zero-intercept slope of 1 ($100 \times$ (zero-intercept slope$-1$)). The mean relative bias is ($100 \times$ (HeiPro$-$X)/X), where X is Pandora-DS, TROPOMI, or GEM-MACH.

**3.1 HeiPro vs. Pandora-DS**

Figure 3a shows the correlation between the HeiPro partial columns and Pandora-DS tropospheric columns. This comparison provides a good opportunity to study the differences between scattered sunlight and direct-Sun measurements by the same instrument. Although the direct-Sun and MAX-DOAS retrieval wavelength ranges are different due to the varying

standard protocols for each, it is worthwhile to note that the HeiPro long vis versus long UV $NO_2$ partial column comparisons showed good agreement with one another, with a zero-intercept slope of $0.97 \pm 0.004$ and mean relative bias of $0.7\% \pm 5.9\%$ (results not shown). We therefore do not expect the choice of retrieval window to significantly impact the HeiPro long UV

partial column comparisons to Pandora-DS (see Table A1 for the HeiPro long vis partial column $NO_2$ comparisons). We find that HeiPro partial columns, although measuring a smaller altitude range, are larger than Pandora-DS tropospheric columns, with a multiplicative bias of $51\% \pm 0.8\%$ and a mean relative bias of $61\% \pm 9.7\%$. Additionally, the multi-year Pandora dataset presented here provides an opportunity to investigate seasonal differences between HeiPro and Pandora-DS. Monthly and hourly box-and-whisker plots are displayed in Fig. 4a and b, respectively. The top and bottom of each box represent the

$75^{th}$ and $25^{th}$ percentiles, respectively, the horizontal line within is the median value, and the whiskers are the most extreme non-outlier data points, where outliers are values that are 1.5 times greater or less than the interquartile range. The mean relative bias between the datasets for each month of the year or hour of the day are depicted by the circled markers. Both HeiPro (0–4 km) and Pandora-DS tropospheric column median values display typical seasonal and diurnal trends for tropospheric $NO_2$, whereby $NO_2$ is greater during the winter months and morning hours from 6–9 a.m. LT due to various chemical and dynamical

factors such as increased emissions, shallow PBL, and photochemistry. The HeiPro partial columns have even larger seasonal and diurnal trends compared to Pandora-DS tropospheric columns. This is further exemplified by the mean relative bias between the datasets, which exhibits a strong seasonal variability, with the largest values observed in the winter (84–114%) and some of the smallest values observed in the summer (39–51%). The larger relative bias during the winter months is also shown by a larger zero-intercept slope value during the winter ($1.60 \pm 0.02$) compared to summer ($1.38 \pm 0.02$), as well as a

greater York linear fit intercept value during the winter ($2.17 \times 10^{15}$ molec cm$^{-2}$) compared to summer ($0.38 \times 10^{15}$ molec cm$^{-2}$). The mean relative bias between the two datasets is also larger from 6–9 a.m. LT (77–131%) compared to 4–7 p.m. LT (32–51%).

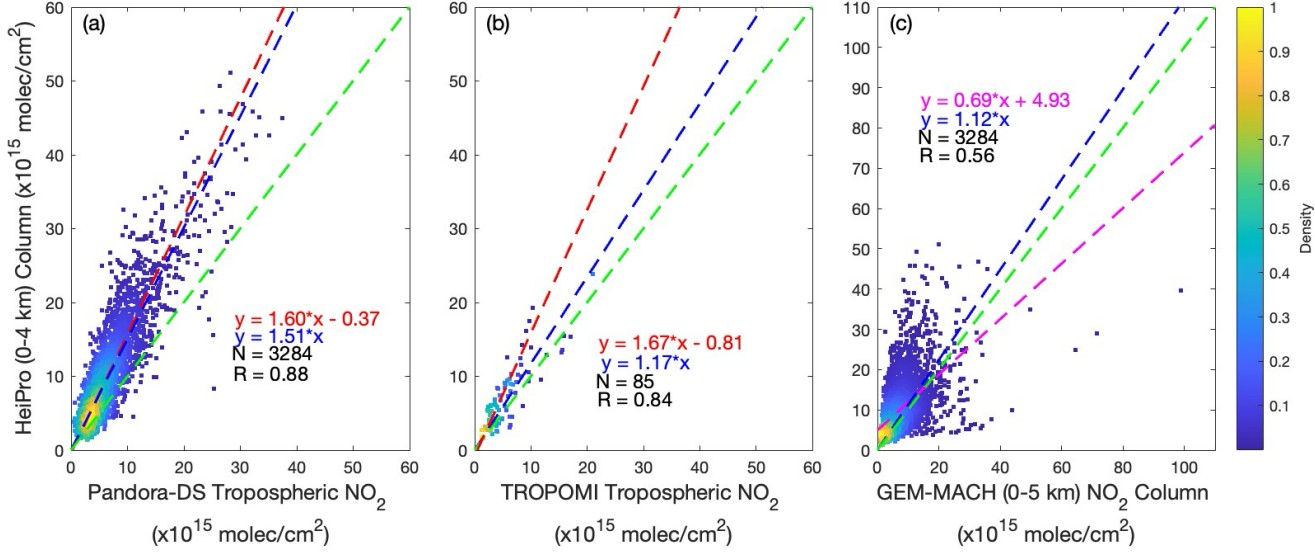

**Figure 3: Comparisons between HeiPro (0–4 km) NO₂ partial columns (2018–2020) vs. (a) Pandora-DS tropospheric columns, (b) TROPOMI tropospheric columns, and (c) GEM-MACH (0–5 km) partial columns. The York linear regression (dashed red line), zero-intercept linear regression (dashed blue line), ordinary least squares regression (dashed magenta line), and the 1:1 line (dashed green line) are depicted. The color bar indicates the normalized density of the data points.**

For reference, HeiPro (0–4 km) vs. Pandora-DS total columns are compared in panel (a) of Fig. A4, which shows that HeiPro partial columns exhibit a positive multiplicative bias of 16% ± 0.7% and a mean relative bias of 6.1% ± 4.8%. Not surprisingly, there is better agreement here as compared to Fig. 3a (i.e., HeiPro vs. Pandora-DS tropospheric $NO_2$) since the Pandora-DS total columns are larger. The TROPOMI vs. Pandora-DS $NO_2$ total and tropospheric column comparisons are shown in Fig. A4b–c, respectively. Pandora-DS and TROPOMI show good agreement with one another for both total column (multiplicative bias: −12% ± 1.9%; mean relative bias: 0.1% ± 21%) and tropospheric $NO_2$ (multiplicative bias: −4.4% ± 3.5% ; mean relative bias: −0.9% ± 34%). Note that the large uncertainties are due to the relatively large TROPOMI total column and tropospheric $NO_2$ errors. Additionally, the tropospheric $NO_2$ agreement in panel (c) provides more confidence in the stratospheric-tropospheric separation method that was used in the study (i.e., Pratmo-OMI data). TROPOMI total column $NO_2$ at this measurement site has been studied and validated in Zhao et al. (2020). Using the version 1 data product, Zhao et al. (2020) found that TROPOMI vs. Pandora-DS total column $NO_2$ had a zero-intercept slope of 0.70 and correlation coefficient of 0.75. The version 2.3 data product used in this work showed an improvement from version 1, with a zero-intercept slope of 0.89 and correlation coefficient of 0.81. The time period of the study in which version 1 was used (March 2018 to March 2019) was similar to that of this study (May 2018 to June 2020). Comparisons and validation of the newer version 2.3 TROPOMI data products are outside the scope of this work.

Next, we explore various contributing factors to the generally large bias of HeiPro to Pandora-DS. We investigate how the PBL height contributes to the bias because the Pandora-DS measurements are missing the first ~15 m of the vertical column due to the rooftop instrument location while the HeiPro measurements attempt to include the first 15 m since the multi-axis viewing geometry has some sensitivity to lower layers and the HeiPro algorithm extrapolates the profile to ground level. This difference in detection may further be amplified by shallower PBL heights during winter months and morning hours when the PBL height is smaller due to lower surface temperatures and less boundary layer dynamics. This leads to less vertical mixing of pollutants, with $NO_2$ accumulating near the surface (e.g., Lin and McElroy, 2010; Chan et al., 2018; Schreier et al., 2019) where it is not captured by the Pandora-DS measurements in the first 15 m. We additionally investigate the effects of the SAA on the bias due to the spatiotemporal heterogeneity of $NO_2$ as well as the varying airmasses measured by the multi-axis and direct-Sun viewing geometries. Lastly, while $NO_2$ increases during the wintertime due to greater anthropogenic emissions from heat sources (e.g., Meng et al., 2018) and increased lifetimes due to decreased solar radiation, it is possible

that increased emissions can contribute to the bias if there are more NO₂ emissions coming from the multi-axis azimuth viewing direction of 255° compared to the various direct-Sun viewing angles.

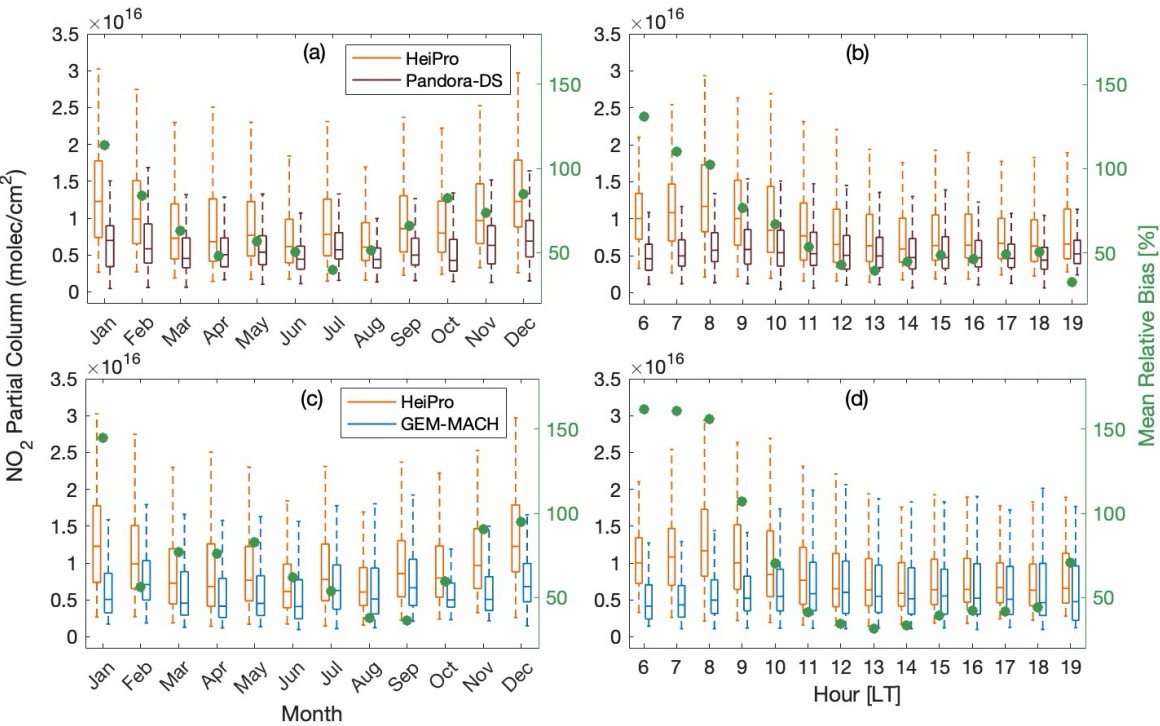

**Figure 4: (a) Monthly and (b) hourly box-and-whisker plots of HeiPro 0–4 km (orange) and Pandora-DS tropospheric (brown) NO₂ columns as well as the mean relative bias between the two (green circles). (c) Monthly and (d) hourly box-and-whisker plots of HeiPro 0–4 km (orange) and GEM-MACH 0–5 km (blue) NO₂ columns as well as mean relative bias between the two (green circles). The top and bottom of each box represent the 75ᵗʰ and 25ᵗʰ percentiles, respectively, the horizontal line within is the median value, and the whiskers are the most extreme non-outlier data points, where outliers are values that are 1.5 times greater or less than the interquartile range.**

The impact of the above factors on the bias is illustrated in Fig. 5a and b, which shows the zero-intercept slope, i.e., multiplicative bias, and mean relative bias, respectively, for HeiPro partial versus Pandora-DS tropospheric NO₂ columns under the following scenarios versus the SAA range, with 20° bins per SAA range: (1) HeiPro partial vs. Pandora-DS standard tropospheric NO₂ columns, (2) HeiPro partial vs. Pandora-DS 'modified' tropospheric NO₂ columns, whereby a 0–15 m column is added to the Pandora-DS column, see Section 3.1.1 for further details, (3) HeiPro partial vs. Pandora-DS modified tropospheric NO₂ columns during summer and winter, (4) HeiPro partial vs. Pandora-DS standard tropospheric NO₂ during winter only. In Sections 3.1.1–3.2.2, we quantify the contributions to the multiplicative and mean relative biases (see markers in Fig. 5c and d, respectively) between HeiPro and Pandora-DS NO₂ from various factors and indicate the scenarios that lead

to the best and worst agreement between the two at this site. The relative contribution is calculated as the absolute value of the percent change in the bias after incorporating a factor (e.g., the PBL contribution is calculated as: $100 \times (| \text{bias of HeiPro vs. Pandora-DS modified} - \text{bias of HeiPro vs. Pandora-DS standard}|)/ \text{bias of HeiPro vs. Pandora-DS standard}$, at constant SAA range), and so any decreases in the bias when incorporating a factor such as PBL are reported as positive numbers and therefore as contributions to the bias.

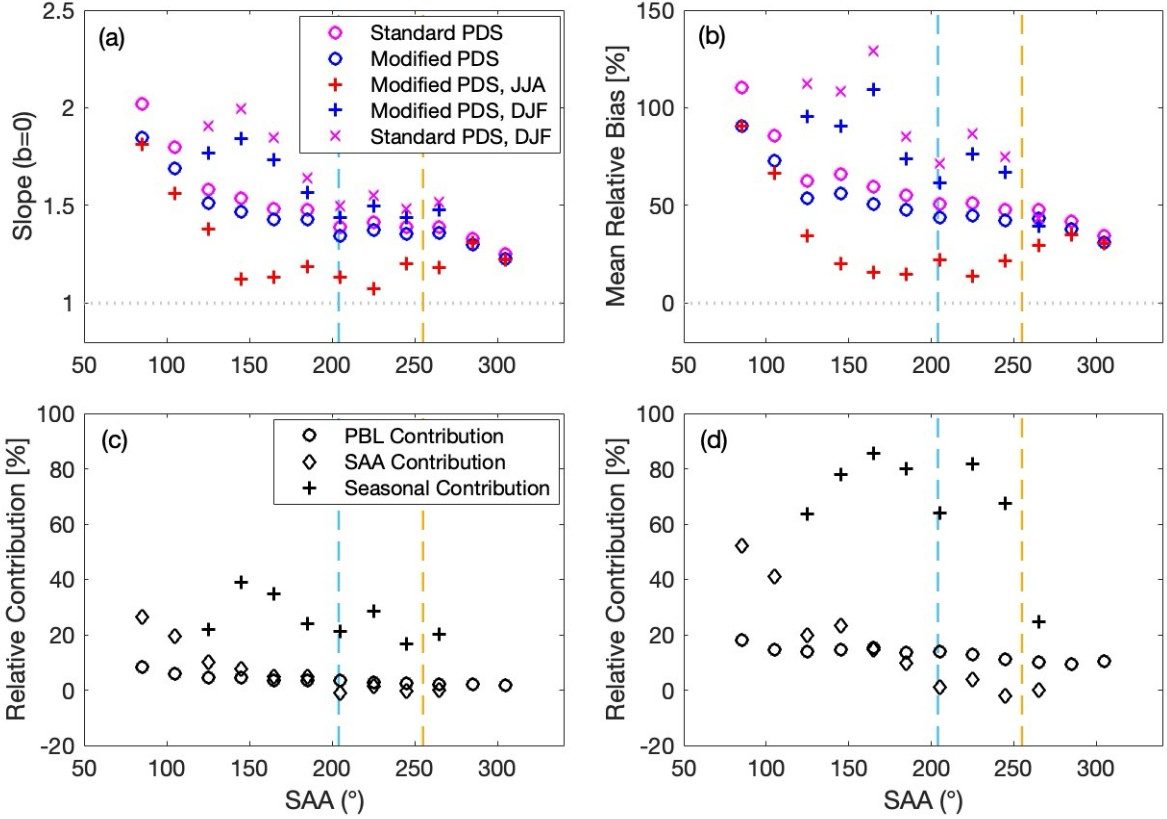

**Figure 5: (a) Zero-intercept slope and (b) mean relative bias for HeiPro partial vs. Pandora-DS (PDS) tropospheric NO$_2$ columns under various scenarios (see legend), with relative contributions (%) from various factors (see legend) to the (c) zero-intercept slope and (d) mean relative bias. The dashed yellow lines represent the Pandora azimuth viewing angle of 255° and the dashed blue lines represent the mean SAA during the TROPOMI overpass time of 204°.**
**Each marker represents a 20° bin of SAA, with a range of values that precede and include the marker value.**

### 3.1.1 PBL height effects

      The effect of the missing 0–15 m in the Pandora-DS tropospheric column can be enhanced when the PBL height is shallow. Overall, we found that this partially contributed to why the HeiPro partial NO$_2$ columns were larger than the Pandora-DS tropospheric columns but was not able to fully account for the biases observed. This was investigated by producing an

approximate 0–15 m column, which was calculated using the NAPS in situ surface NO₂ value taken as an indication of the first 15 m of the NO₂ profile at the measurement location, calculated as:

$$0 - 15m\ column = \frac{h \times VMR \times n_A \times P}{R \times T \times 10^9\ molec.\ air} \qquad Equation\ (1)$$

where $h$ is the height of the column (15 m), $VMR$ is the volume mixing ratio of NO₂ in ppbv $\left(\frac{X\ molecules\ of\ NO_2}{10^9\ molecules\ of\ air}\right)$, $n_A$ is Avogadro's number, $P$ is the surface pressure in $atm$, $R$ is the ideal gas constant ($8.21 \times 10^{-5} m^3 \cdot atm \cdot K^{-1} \cdot mol^{-1}$), and $T$ is the surface temperature in $K$. Surface pressure and temperature values were obtained from ERA5 reanalysis data for each hour corresponding to the in situ measurement of NO₂ VMR. This column was then added to the Pandora-DS tropospheric column to create a 'modified' column and the biases between HeiPro and Pandora-DS were re-examined using the modified column. The effects of the missing 15 m and the PBL height on the bias can be quantified by looking at how the multiplicative bias and mean relative bias change from the HeiPro partial versus Pandora-DS standard tropospheric columns to the HeiPro partial versus Pandora-DS modified tropospheric columns. The most notable reductions in the biases are seen when the SAA range is 66°–85°, although large multiplicative biases and mean relative differences of > 50% remain at this SAA range. Such reductions in the multiplicative and relative biases (HeiPro partial column vs. Pandora-DS tropospheric column − HeiPro partial column vs. Pandora-DS modified tropospheric column) are depicted by the circles in Fig. 5c and d and termed the "PBL contribution", respectively. The largest reductions of 8.4% in the multiplicative bias and 18% in the mean relative bias occur at the SAA range of 66°–85°, which corresponds to the morning hours of 6–8 a.m. LT. This is unsurprising since the PBL height is shallow during this time, which presents a greater opportunity for any NO₂ accumulating in the first 15 m to be missed by the Pandora-DS measurements. The impact of accounting for the missing 15 m then drops to negligible amounts throughout the day as the SAA increases, presumably due to a more well-mixed PBL. Because the 0–15 m column derivation assumes a constant NO₂ VMR from 0 to 15 m, the estimations provided here of the PBL contribution to the bias represent upper limits (aside from cases with lofted plumes below 15 m). In cases where the NO₂ VMR at 15 m < NO₂ VMR at 0 m, the contribution of the 0–15 m column would be lower, and therefore, the PBL contribution to the bias would be lower.

**3.1.2 NO₂ heterogeneity contribution**

Another potential factor contributing to the large bias of HeiPro to Pandora-DS is the three-dimensional heterogeneity of the NO₂ field combined with the fact that multi-axis and direct-Sun viewing geometries inherently measure different airmasses. Importantly, the direct-Sun and multi-axis viewing geometries point to different azimuth angles, with direct-Sun tracking the Sun and multi-axis constantly pointing to 255°, which also contributes to differences in the airmasses being probed due to the spatial heterogeneity of NO₂. To investigate how differences in viewing angles contributes to the bias, the zero-intercept slope and mean relative bias between HeiPro partial NO₂ columns versus Pandora-DS modified tropospheric NO₂

columns at each SAA range were compared to the bias at 246°–265°, i.e., Pandora-DS measurements within $\pm10°$ of the MAX-DOAS HeiPro measurements. This bias difference is represented by the diamond markers in Fig. 5c and d and termed the "SAA contribution". The SAA contribution was not calculated for the SAA ranges of 266°–285° since the HeiPro bias relative to Pandora-DS in these ranges is smaller than the bias at the reference SAA range of 246°–265°.

The largest zero-intercept slope and mean relative biases (1.85 and 91%, respectively) are seen when the direct-Sun and multi-axis viewing angles are farthest apart during the measurement day in the hours from 6–8 a.m. LT. The zero-intercept slope then drops from 1.85 at the 66°–85° SAA range to 1.36 at the 246°–265° SAA range, while the mean relative bias drops from 91% to 44% for these ranges, representing 27% and 52% reductions in the multiplicative and mean relative biases, respectively. These large contributions are during the morning hours when the direct-Sun and multi-axis viewing geometries are farther apart, and the SAA contribution to the bias decreases to negligible values as the SAA increases. It is important to note that even when Pandora-DS measurements are within $\pm10°$ of the HeiPro measurements, the biases are still large, with a zero-intercept slope of 1.30 and mean relative bias of 44%. One contribution to this bias may be the 10° differences between direct-Sun and multi-axis viewing angles, which may still exhibit $NO_2$ heterogeneity, as well as the different horizontal sensitivities between the direct-Sun and multi-axis viewing geometries.

Additionally, since the mean relative bias between HeiPro partial versus Pandora-DS tropospheric columns was at a maximum during winter months and minimum during summer months (see Fig. 4a), seasonal effects on the bias were investigated. The zero-intercept slope and mean relative bias between HeiPro partial versus Pandora-DS modified tropospheric columns were plotted against the SAA range for summer months (JJA) and winter months (DJF, see Fig. 5a and b), and are both larger during the winter months compared to summer months across all SAA ranges, which is consistent with the mean relative bias shown in Fig. 4a. The differences in the biases of HeiPro partial versus Pandora-DS tropospheric columns between winter and summer at each SAA range are depicted by the crosses in Fig. 5c and d and termed the "seasonal contribution". Since there are no Pandora wintertime measurements for the first two SAA ranges and the last SAA range, seasonal contributions cannot be assessed for those hours of the day (at 6 a.m. and onwards from 6 p.m., LT). The seasonal contribution to (i) the zero-intercept slope ranges from 16% to 39% and to (ii) the mean relative bias ranges from 25% to 85%, depending on the SAA range. The seasonal contribution peaks when the SAA range is between 126°–165°, and then decreases as the SAA increases. Due to increased $NO_2$ emissions during the winter, it is possible that the seasonal contribution stems from increased emissions from the multi-axis azimuth viewing direction of 255°, and so HeiPro measurements, and subsequently the biases towards Pandora-DS, are even larger during winter months.

Next, we investigate the spatiotemporal $NO_2$ heterogeneity around Downsview and assess whether we can observe the heterogeneity that supports the biases observed (i.e., large differences in the $NO_2$ columns between the azimuth viewing angle and the various SAAs, and even larger differences between the two from 6–9 a.m. LT and winter months). To do this,

470 the NO$_2$ field around Downsview was explored using two datasets from 2018–2020: (i) TROPOMI tropospheric NO$_2$ columns and (ii) GEM-MACH (0–5 km) NO$_2$ columns. The TROPOMI pixel-averaged (Fioletov et al., 2011; Sun et al., 2018) NO$_2$ field was used to investigate the larger bias in winter, while the mean GEM-MACH NO$_2$ field was used to investigate the larger bias in the early morning hours since TROPOMI only provides data at 13:30 LST. We found that using such datasets to demonstrate that there was NO$_2$ heterogeneity between the multi-axis and direct-Sun viewing directions was insufficient

475 because (i) TROPOMI tends to underestimate NO$_2$ in polluted regions and is not as suited to capture small local NO$_2$ enhancements as MAX-DOAS measurements are (Verhoelst et al., 2021, see Section 3.2 for further discussion), and (ii) the GEM-MACH model resolution of 10 km × 10 km may also not capture local enhancements and it is difficult to use model inventories as an interpretation of actual conditions since the model utilizes inventories from the year 2013. See Fig. A5 and A6 in the Appendix for the TROPOMI and GEM-MACH NO$_2$ fields, respectively, and for further discussion.

   To summarize, the smallest biases in the dataset are observed during the summer months, when using Pandora-DS modified tropospheric columns and when the SAA > 125°, which exhibit multiplicative biases of 7.2% ± 3.4% to 31% ± 2.0% and mean relative biases of 14% ± 9.0% to 35% ± 11%. We term these combined conditions the "best-case scenario", which constitutes the following: a well-mixed PBL, accounting for the missing 15 m, smaller viewing angle differences, and

485 summer months with less NO$_2$ emissions. Therefore, when conditions are suitable for agreement, HeiPro partial columns show good agreement with Pandora-DS tropospheric NO$_2$ columns. Conversely, the largest biases are observed when using Pandora-DS standard columns (i) during the winter months throughout the measurement day (see HeiPro partial column bias to Pandora-DS standard columns in Fig. 5a and b), which exhibit multiplicative and mean relative biases mostly greater than 50% and 70%, respectively, and (ii) during the early morning hours across all seasons (see HeiPro partial column bias to Pandora-DS

490 standard columns when the SAA ≤ 105° in Fig. 5a and b), which exhibit multiplicative biases of 80% ± 2.6% to 102% ± 3.1% and mean relative biases of 86% ± 10% to 110% ± 12%. We term the conditions outlined in (i) and (ii) as the "worst-case scenario". Therefore, some portion of the large bias of HeiPro to Pandora-DS may be explained by the following environmental conditions: shallow PBL, the missing 15 m in the Pandora-DS measurements, larger viewing angle differences, and increased NO$_2$ heterogeneity during the winter. Scatter plots for the best-case (multiplicative bias: 19% ± 1.3%; mean

495 relative bias: 24% ± 9.7%) and worst-case (multiplicative bias: 72% ± 1.7%; mean relative bias: 101% ± 9.4%) subsets of the data are presented in Fig. A7a and b, respectively. Although the biases are minimized in the best-case scenario, nonzero biases remain. This is shown in Fig. 5a and b by the offset from the dotted grey line of the HeiPro partial column bias towards Pandora-DS modified tropospheric columns during the summer months when the SAA > 125°). Possible contributing factors to the remaining bias are discussed in Section 5.

## 3.2 HeiPro vs. TROPOMI

   Figure 3b shows the scatter plot for HeiPro partial versus TROPOMI tropospheric NO$_2$ columns, with HeiPro exhibiting multiplicative and mean relative biases of 17% ± 4.0% and 37% ± 51%, respectively, compared to TROPOMI

measurements. The relatively large uncertainty in the mean relative bias can be attributed to the larger retrieval errors in the TROPOMI tropospheric $NO_2$ measurements. Our findings regarding the differences between HeiPro and TROPOMI are in accordance with a study by Verhoelst et al. (2021), who categorized the TROPOMI bias relative to ground-based MAX-DOAS $NO_2$ into the following regimes: (1) regions with low pollution levels can have median relative differences ($100 \times$ (TROPOMI−MAX-DOAS)/MAX-DOAS) of up to −27%, (2) regions with moderate pollution levels exhibit biases between −15% to −56%, and (3) for extremely polluted regions, differences of −37% to −74% are seen. The pollution level categories were based on the median MAX-DOAS ground-based tropospheric columns. As Toronto is a moderately polluted region with a median HeiPro 0–4 km partial column of $7.71 \times 10^{15}$ molec $cm^{-2}$, our findings are in accordance with the bias range in the moderately polluted categorization of Verhoelst et al. (2021), i.e., the median bias (TROPOMI−HeiPro) within this dataset is $-1.4 \times 10^{15}$ molec $cm^{-2}$, with a median relative bias of −27%, as per the equation used in the study. Factors contributing to the discrepancy between TROPOMI and MAX-DOAS $NO_2$, and therefore also the limitations of using the TROPOMI $NO_2$ field to address the HeiPro overestimation of Pandora-DS, are discussed next.

Firstly, MAX-DOAS measurements can capture more local enhancements in $NO_2$, while the satellite retrievals provide a smoothed pixel representation. For example, satellite underestimation of MAX-DOAS measurements can occur if the footprint of the emission source is smaller than the satellite footprint of 3.5 km × 5.5 km for TROPOMI (3.5 km × 7.5 km prior to August 2019, Verhoelst et al., 2021). Additionally, the TROPOMI retrieval algorithm is sensitive to the a priori $NO_2$ profile shape, and the use of low-resolution a priori $NO_2$ profiles in the TROPOMI retrieval algorithm can contribute to the underestimation of ground-based MAX-DOAS measurements, as shown in various studies (Zhao et al., 2020; Dimitropoulou et al., 2020). For example, Zhao et al. (2020) showed a 10% reduction in the bias between TROPOMI and Pandora-DS $NO_2$ total columns when replacing the standard a priori profile in the TROPOMI retrieval algorithm with GEM-MACH, a profile from a high-resolution regional air quality forecast model.

The TROPOMI overpass time of 13:30 LST occurs at a time of day such that the HeiPro biases relative to Pandora-DS are at the lower end of the bias range (see where markers intersect with the dashed line at 204° in Fig. 5a and b). It would be interesting to observe how the bias of geostationary $NO_2$ measurements (e.g., TEMPO) relative to ground-based direct-Sun and multi-axis measurements changes throughout the day, given the large seasonal and diurnal dependency of the HeiPro bias relative to Pandora-DS. The results of the HeiPro comparisons to Pandora-DS in this study can therefore aid in future ground-based direct-Sun and MAX-DOAS validation studies of TEMPO by providing possible explanations for differences in the bias between TEMPO versus MAX-DOAS and TEMPO versus direct-Sun at this measurement site.

### 3.3 HeiPro vs. GEM-MACH

Here we compare the HeiPro $NO_2$ partial columns to GEM-MACH. Figure 3c shows the regression between HeiPro (0–4 km) and GEM-MACH (0–5 km) partial columns. While the GEM-MACH standard dataset includes partial columns from 0–5 km, we did not generate a 0–4 km partial column because we found the GEM-MACH $NO_2$ VMRs from 4–5 km to be very small, and that the integrated $NO_2$ from 4–5 km constituted only 0.3% of the GEM-MACH (0–5 km) partial columns. HeiPro is greater than GEM-MACH with a multiplicative bias of 12% $\pm$ 1.2% and a mean relative bias of 67% $\pm$ 7.1%. Figure 4c

and d display monthly and hourly, respectively, box-and-whisker plots of both datasets as well as the mean relative bias between them. As with comparisons of HeiPro to Pandora-DS, the HeiPro dataset exhibits a stronger annual cycle than GEM-MACH, with a larger interquartile range and whiskers during the winter months. The seasonal trend appears weaker in the GEM-MACH box-and-whisker plots, which do not seem to capture the seasonal pattern in $NO_2$. Such differences in seasonal patterns between GEM-MACH and HeiPro are further reflected in the monthly mean relative biases (see circled markers), in

which the winter months exhibit some of the largest percent differences (95–145%) compared to the summer months (38–53%). Also similar to the HeiPro and Pandora-DS hourly comparisons, the HeiPro versus GEM-MACH hourly comparisons exhibit the largest (107–162%) mean relative biases during the morning hours (6–9 a.m. LT), with values generally decreasing throughout the day. The large bias of HeiPro to GEM-MACH $NO_2$ can partially be explained by the $NO_2$ inventories used in the GEM-MACH model. The model inventories account for surface $NO_2$ emissions in the PBL, but do not include lightning

and aircraft emissions of $NO_2$ in the free troposphere. This is further demonstrated in the $NO_2$ profile and surface comparisons in Section 4.

Due to the similarly large biases in HeiPro partial columns relative to Pandora-DS and to GEM-MACH, as well as similar seasonal and diurnal patterns in these biases, a comparison was done between Pandora-DS and GEM-MACH (see

Appendix, Fig. A8). The scatter plot (Fig. A8a) shows a zero-intercept slope of 1.00 (i.e., no apparent multiplicative bias), indicating that there is good agreement between Pandora-DS tropospheric and GEM-MACH (0–5 km) partial $NO_2$ columns, although a positive mean relative bias ($100 \times$ (GEM-MACH$-$ Pandora-DS)/Pandora-DS) of 20% exists. The positive mean relative bias, indicating that GEM-MACH values are greater than Pandora-DS, is also evident in the mean monthly and hourly box-and-whisker plots in Fig. A8b and c, respectively, which show that GEM-MACH is greater than Pandora-DS, particularly

in the afternoon hours with an approximate mean relative bias of 30%.

### 4 HeiPro surface $NO_2$ & profile comparisons

This section presents the surface $NO_2$ comparisons between HeiPro versus (i) NAPS in situ and (ii) GEM-MACH, as well as $NO_2$ profile comparisons between HeiPro and GEM-MACH. The surface comparisons are presented in Section 4.1 while the profile comparisons are presented in Section 4.2. GEM-MACH is used for the profile comparison as it is the only

source of $NO_2$ profile data available at this site. The multiplicative bias is as described in Section 3 and the mean relative bias is ($100 \times (HeiPro - X)/X$), where X is NAPS in situ or GEM-MACH.

## 4.1 Surface $NO_2$ comparisons

Figure 6 shows the scatter plots of HeiPro versus (a) NAPS in situ surface $NO_2$ and (b) GEM-MACH surface $NO_2$, whereby HeiPro surface $NO_2$ exhibits negative multiplicative biases of $-5.8\% \pm 0.7\%$ and $-41\% \pm 0.5\%$ to NAPS in situ
and GEM-MACH, respectively. Likewise, HeiPro exhibits mean relative biases of $-9.7\% \pm 7.5\%$ and $-37\% \pm 2.4\%$ to NAPS in situ and GEM-MACH, respectively. Figure 6c–f shows box-and-whisker plots of the seasonal diurnal trends of HeiPro, GEM-MACH, and NAPS in situ surface $NO_2$ (box-and-whisker plots are as defined in Section 3.1). All datasets display peaks in surface $NO_2$ around 6–8 a.m. LT and decrease throughout the day, capturing the diurnal trend of surface $NO_2$, which is shaped by emission intensity, PBL dynamics, and photochemistry. Additionally, the small negative bias between
HeiPro and NAPS in situ is evident in the diurnal plots throughout most of the measurement day and across all seasons. On the other hand, the larger negative bias of HeiPro to GEM-MACH is evident in the diurnal plots throughout the measurement day, particularly during the morning hours.

Across all seasons, the largest surface $NO_2$ values are from the GEM-MACH data from 6–9 a.m. LT. The GEM-
MACH surface $NO_2$ is likely to be overestimated due to an outdated $NO_2$ inventory being used in the model, which does not account for reduced $NO_2$ emissions over the years or reduced commutes during the COVID-19 lockdown periods in 2020 (Zhao et al., 2022). Additionally, all three datasets capture a stronger diurnal trend on weekdays (Fig. 6g) compared to the diurnal trend on weekends (Fig. 6h), which shows smaller median values as well as interquartile and whisker ranges. It is important to note that the HeiPro seasonal diurnal surface $NO_2$ values, as well as weekday versus weekend trends, are more
closely aligned to NAPS in situ surface $NO_2$ than GEM-MACH is to NAPS in situ surface $NO_2$. HeiPro provides a reasonable estimate of surface $NO_2$ values as well as the seasonal diurnal pattern of surface $NO_2$ but tends to underestimate the intensity during the evening hours. Contributions to the discrepancy between HeiPro and NAPS in situ surface $NO_2$ are discussed next.

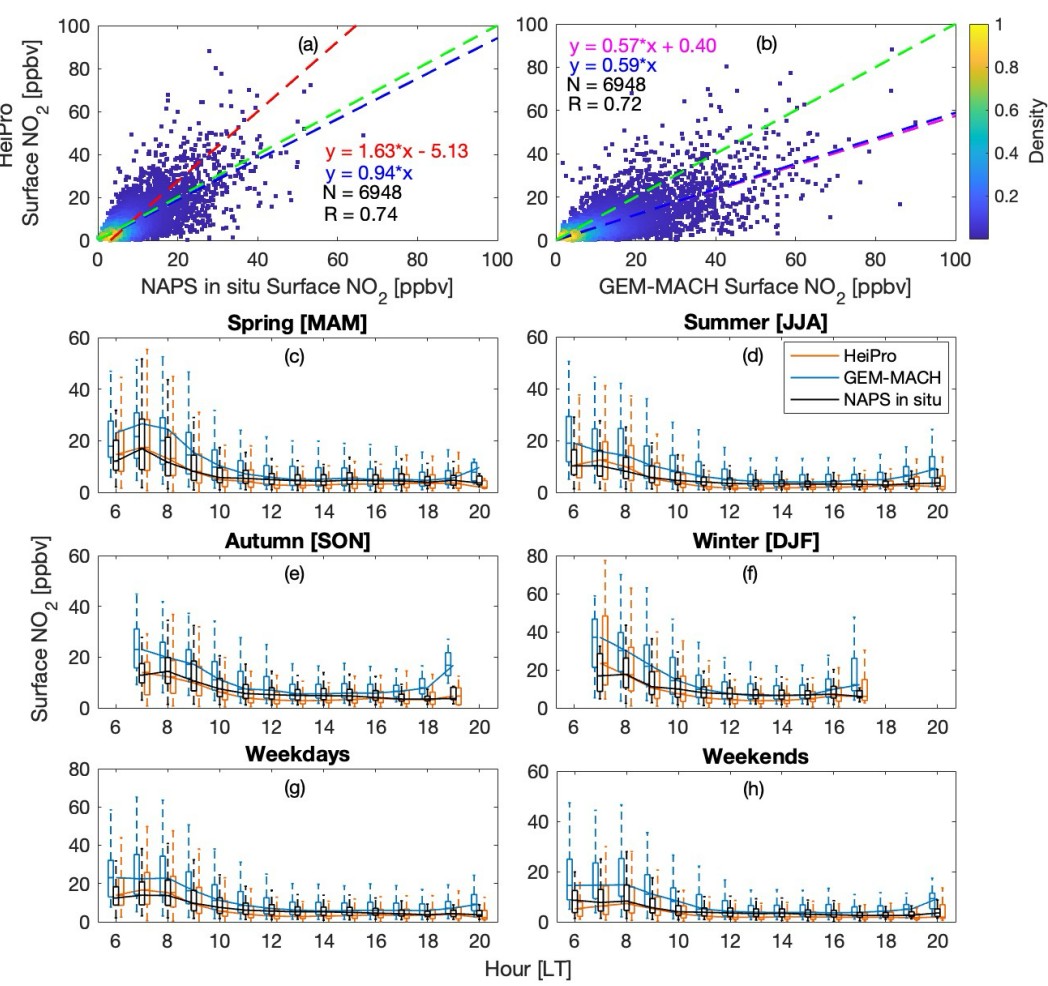

**Figure 6: Comparisons of surface NO₂ VMR (2018–2020) for HeiPro vs. (a) NAPS in situ, and (b) GEM-MACH. The dashed lines and color bar are as indicated in Fig. 3. (c)–(f) Seasonal diurnal and (e), (f) weekday, weekend box-and-whisker plots of HeiPro (orange), GEM-MACH (blue), and NAPS in situ (black) surface NO₂ VMRs. Box-and whisker values are as defined in Fig. 4.**

Due to the vertical offset of 11 m between the Pandora and in situ instruments (the former on the rooftop at 15 m and the latter at 4 m above ground level) and the heterogeneity of the NO₂ field, PBL height was investigated as a potential source of discrepancy between HeiPro and NAPS in situ surface NO₂. The relative bias of HeiPro to NAPS in situ ($100 \times$ (HeiPro−NAPS in situ)/NAPS in situ) versus PBL height range is presented in the box-and-whisker plots of Fig. 7. The box-and-whisker values are as defined in Section 3.1. The PBL range were chosen to ensure that the number of data points in each bin were in the same order of magnitude. For the shallowest PBL range from 0–0.40 km, the upper and lower extents of the boxes and whiskers indicate that, at times, HeiPro is greater than NAPS in situ surface NO₂, while at other times, HeiPro is

smaller than NAPS in situ. Although the HeiPro profiles have some sensitivity to the 15 m of the atmospheric column below the instrument altitude, the vertical offset of the Pandora instrument may impact its sensitivity to those 15 m. It is not known how the HeiPro surface value would change if the instrument were on ground level or had negative elevation viewing angles (and therefore more sensitivity to the first 15 m). Nonetheless, it appears that the shallowest PBL range exhibits a larger range

of differences but smaller (less negative) median values (e.g., for summer, whisker range: $-128\%$ to $195\%$; median value: $-10\%$) compared to the largest PBL range (e.g., for summer, whisker range: $-119\%$ to $22\%$; median value: $-51\%$), and so the vertical offset between the instruments may contribute to the larger range of differences observed when the PBL height is shallow and there is more vertical $NO_2$ heterogeneity. In addition to the narrower whisker range for PBL heights $> 0.90$ km, this PBL range consistently also has median values of approximately $-50\%$ across all seasons (see horizontal lines within

each box in Fig. 7). At this PBL range, which corresponds to the afternoon periods when surface $NO_2$ is at a minimum, the airmass measured by both the Pandora and NAPS instruments is more uniform, which may contribute to a more consistent difference between the two, hence the narrower whisker range. Additionally, the NAPS instrument measures a very localized airmass due to the nature of the in situ technique, while the HeiPro measurements stem from an instrument field-of-view of $1.6°$ (Herman et al., 2009) and a photon effective pathlength between 5 and 10 km for UV measurements (Ortega et al., 2015).

These varying horizontal sensitivities between HeiPro and NAPS in situ, combined with spatiotemporal $NO_2$ heterogeneity, can also contribute to differences in the airmasses being measured.

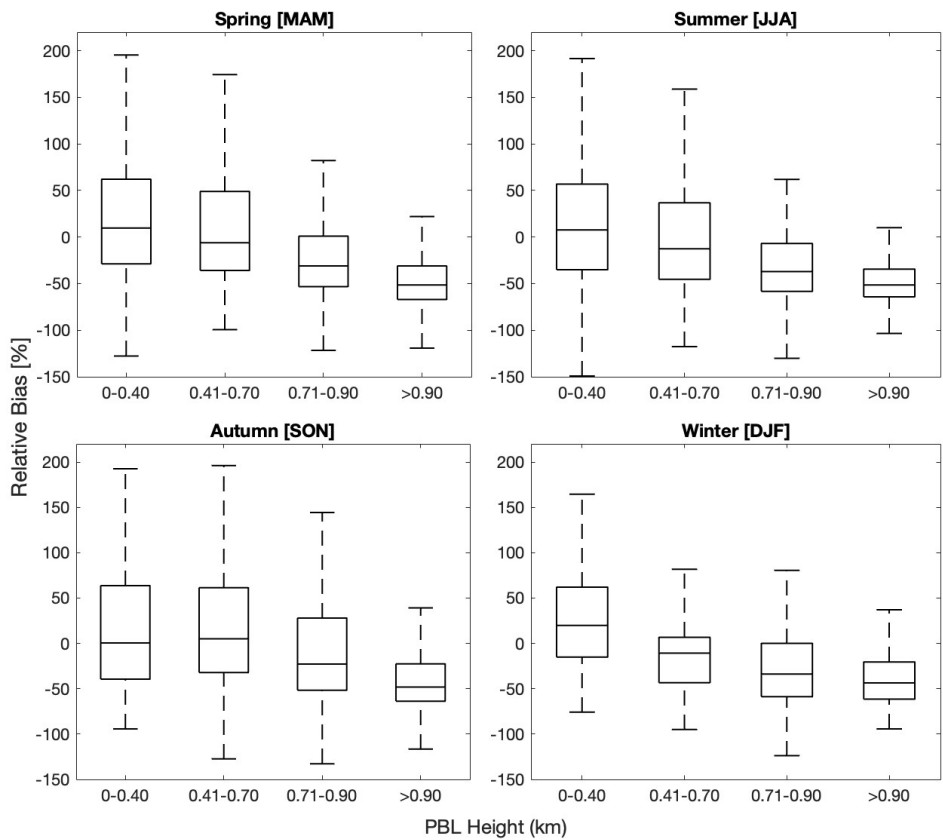

**Figure 7: Seasonal box-and-whisker plots of the relative bias of HeiPro to NAPS in situ surface NO₂ vs. PBL height range. Box-and-whisker values are as defined in Fig. 4.**

## 4.2 Profile comparisons

Figure 8 shows the seasonal median NO₂ profiles from HeiPro, GEM-MACH, and GEM-MACH smoothed by the HeiPro averaging kernel. The shaded regions represent the 90[th] percentile (right of solid line) and 10[th] percentile (left of solid line) values for HeiPro and GEM-MACH but are not shown for GEM-MACH smoothed for visual clarity purposes. The seasonal median NAPS in situ surface NO₂ values are also displayed for reference. As stated previously, the surface values for HeiPro and GEM-MACH were extrapolated from the midpoint of the grid level closest to the surface (e.g., 0–200 m for HeiPro), for each available profile. Due to the presence of some lofted layers in the HeiPro profiles, in which the 0–200 m grid level NO₂ VMR is less than that of the 200–400 m grid level, the extrapolated surface VMR for such cases is smaller than that of the grid level closest to the surface. When plotting the median profiles, as is done in Fig. 8, it appears as though the extrapolation to the surface does not have a sharper gradient towards the surface (see HeiPro profiles in Fig. 8), although this is only because the median values are plotted; aside from profiles with lofted layers, each individual profile has a larger increase from the 100 m layer to the surface. HeiPro, NAPS in situ, and GEM-MACH datasets all have larger median surface NO₂

values during the winter months (5.7, 7.8, and 10 ppbv, respectively) compared to spring (4.2, 5.4, and 7.7 ppbv), summer (2.7, 4.0, and 6.5 ppbv), and autumn (4.3, 5.7, and 8.1 ppbv). This is in accordance with winter conditions that increase surface $NO_2$ such as larger anthropogenic emissions, meteorological conditions, etc.

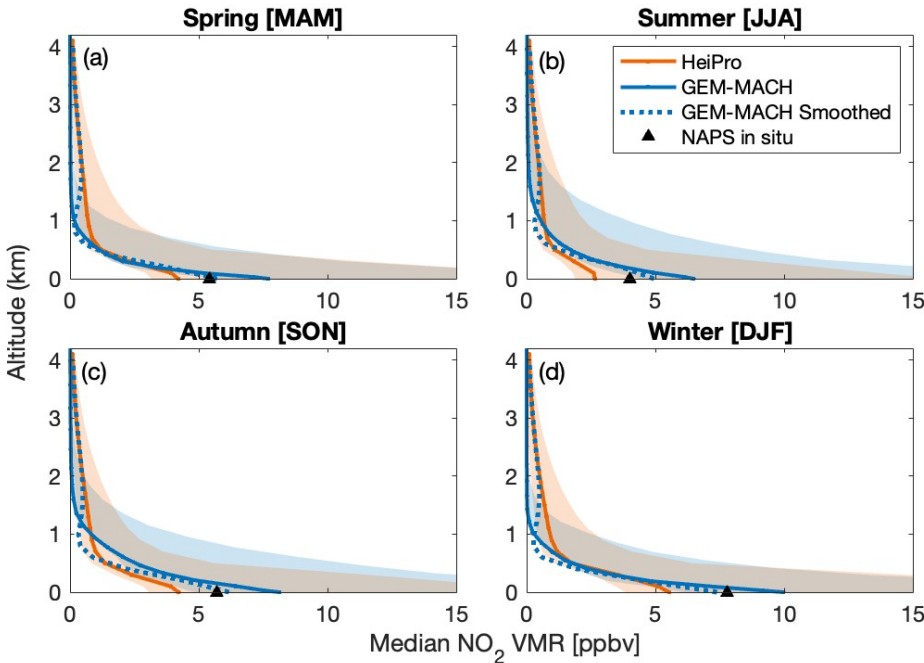

**Figure 8: HeiPro (solid orange line), GEM-MACH (solid blue line), and GEM-MACH smoothed (dotted blue line) median $NO_2$ profiles for each season. The shaded regions represent the 90th percentile (right of line) and 10th percentile (left of line) values for HeiPro and GEM-MACH. The black triangles indicate the seasonal median NAPS in situ surface $NO_2$ values.**

Across all seasons, the HeiPro $NO_2$ median profiles from 0–200 m underestimate the unsmoothed GEM-MACH median values, while from 1.5–4 km, the HeiPro median profiles then overestimate the unsmoothed GEM-MACH median values. For the 0–200 m layer, the mean relative bias of HeiPro towards GEM-MACH decreases from −37% (unsmoothed) to −6.1% (smoothed). Note that these biases are representative of the integrated 0–200 m layer and may differ slightly from the surface values reported in Table 2. The most significant changes occur in the layer from 1.5–4 km, where the HeiPro bias towards GEM-MACH decreases from > 1000% (unsmoothed) to 2.6% (smoothed). The HeiPro surface underestimation and free tropospheric overestimation of the unsmoothed GEM-MACH profiles can probably be explained by the $NO_2$ inventories used in the GEM-MACH model, which, respectively, (i) utilize older inventories that do not account for reduced emissions over the years, and (ii) do not account for free tropospheric $NO_2$ sources while the a priori $NO_2$ profile contains free tropospheric $NO_2$. For the layers above 200 m and below 1 km, there is no clear pattern between HeiPro and GEM-MACH.

Although it is difficult to assess the accuracy of the HeiPro NO$_2$ profiles without the availability of in situ NO$_2$ profiles at this site, it is worthwhile to note that HeiPro seems to provide a conservative estimate of the NAPS in situ value at the surface, where NO$_2$ VMRs are typically the largest. Additionally, while the GEM-MACH surface NO$_2$ VMRs are appropriately larger in the winter, this trend is not similarly captured for the GEM-MACH partial columns (see box-and-whisker plots in Fig. 4c). On the other hand, both the HeiPro surface NO$_2$ and partial columns display increased values during winter months.

## 5 Conclusions

This study produced a three-year dataset of NO$_2$ profiles and partial columns from 0–4 km at Downsview, Ontario by applying HeiPro, an optimal estimation profile retrieval algorithm, to Pandora UV MAX-DOAS measurements. Table 2 provides a summary of all the intercomparisons made using the HeiPro NO$_2$ data products at this measurement site. The HeiPro partial NO$_2$ columns obtained from the profiles were found to overestimate the partial columns from Pandora-DS, TROPOMI, and GEM-MACH, with multiplicative biases of 51%, 17%, and 12%, respectively, and mean relative biases of 61%, 37%, and 67%, respectively. The seasonal and diurnal trends in the mean relative biases between HeiPro and Pandora-DS as well as HeiPro and GEM-MACH were similar, with larger mean relative biases during the winter months and morning hours from 6–8 a.m. LT. Additionally, HeiPro partial columns exhibit larger variability, as evidenced by the larger box-and-whisker ranges in Fig. 4a and b, compared to Pandora-DS; this is consistent with a study by Pinardi et al. (2020), who found that MAX-DOAS measurements tend to depict a fuller range of NO$_2$ variability due to their ability to measure under partially cloudy conditions, while direct-Sun measurements require clear skies. Although Fig. 4 presents hourly-averaged coincident measurements between HeiPro and Pandora-DS, the measurements are not perfectly coincident during that hour that is averaged, which can also contribute to the differences between the direct-Sun and MAX-DOAS datasets. Seasonal and diurnal comparisons to TROPOMI were not possible due to limited datapoints during winter months and there being one measurement per day, respectively. The TROPOMI bias to HeiPro reported in this study (−27%) matched the bias range reported by Verhoelst et al. (2021) for a moderately polluted region, and it would be interesting to investigate how this satellite bias relative to ground-based MAX-DOAS measurements changes throughout the day using a geostationary satellite data product such as that of TEMPO (Zoogman et al., 2017).

**Table 2. Summary of the multiplicative biases and mean relative biases (± uncertainties) for the HeiPro comparisons to partial columns and surface NO$_2$.**

| Datasets compared | Multiplicative Bias (%) | Mean Relative Bias (%) |
|---|---|---|
| HeiPro (0-4 km) vs. Pandora-DS tropospheric NO$_2$ | 51% $\pm$ 0.8% | 61% $\pm$ 9.7% |
| HeiPro (0-4 km) vs. TROPOMI tropospheric NO$_2$ | 17% $\pm$ 4.0% | 37% $\pm$ 51% |
| HeiPro (0-4 km) vs. GEM-MACH (0-5 km) NO$_2$ | 12% $\pm$ 1.2% | 67% $\pm$ 7.1% |

| | | |
|---|---|---|
| **HeiPro vs. NAPS in situ surface NO₂** | $-5.8\% \pm 0.7\%$ | $-9.7\% \pm 7.5\%$ |
| **HeiPro vs. GEM-MACH surface NO₂** | $-41\% \pm 0.5\%$ | $-37\% \pm 2.4\%$ |
| **HeiPro vs. GEM-MACH-smoothed surface NO₂** | $-30\% \pm 0.8\%$ | $-6.0\% \pm 3.4\%$ |

We found that the HeiPro bias relative to Pandora-DS can partially be explained by several factors, which vary in their contributions to the bias throughout the day. The PBL height, combined with the missing 0–15 m partial column in the Pandora-DS measurements, contributed a maximum of 8.4% of the multiplicative bias and 18% of the mean relative bias in the morning hours, with these values declining to < 5% in the evening hours. The differences between the direct-Sun and multi-axis azimuthal viewing angles throughout the day, combined with the spatiotemporal heterogeneity of the NO₂ field, contributed a maximum of 27% and 52% of the multiplicative and mean relative biases, respectively, when the two viewing angles were farthest apart, and declined throughout the day as the direct-Sun azimuthal viewing angle approached the multi-axis one. Lastly, the maximum seasonal contributions to the multiplicative and mean relative biases ranged from 39% and 85% respectively, with systematically larger biases during the winter months. We were not able to assess seasonal contributions at certain SAA ranges during the wintertime since there were no measurements during these hours, i.e., at 6 a.m. and onwards from 6 p.m. LT. We utilize best-case (multiplicative bias: 19%; mean relative bias: 24%) and worst-case (multiplicative bias: 72%; mean relative bias: 101%) scenarios to exemplify how these contributing factors affect the bias, and note that when these factors are minimized, the bias decreases and the agreement between HeiPro and Pandora-DS significantly improves.

There are various possible contributions and sources of uncertainty to the remaining bias. (1) The HeiPro and Pandora-DS comparisons use hourly averages and are not perfectly temporally coincident. This, combined with NO₂ variability, may contribute to the scatter within the data. (2) The HeiPro data stems from NO₂ retrieved in the UV window (338–370 nm) while the Pandora-DS data stems from NO₂ retrieved in the 400–440 nm range, so differences in the spectroscopic analysis may be a factor but we do not expect this to be a large contribution due to the similar agreement of HeiPro long UV vs. long vis partial columns (see Table A1). (3) Uncertainties in the Pratmo model that was used to calculate OMI stratospheric NO₂ throughout the day can overestimate or underestimate Pandora-DS tropospheric NO₂, thereby underestimating or overestimating the bias, respectively. We do not believe that the use of OMI stratospheric NO₂ itself contributed to the large bias, since, for example, TROPOMI stratospheric NO₂ was greater than OMI stratospheric NO₂ at this site. (4) The AOD retrieved from HeiPro is used as a parameter for the radiative transfer model of the trace gas profile retrieval as it helps to constrain the atmospheric light path. Inaccuracies in this retrieved AOD can therefore lead to errors in the NO₂ retrievals. (5) The ERA5 temperature profile data utilized in the HeiPro algorithm may underestimate measurements, which can contribute 3% to the remaining bias (as demonstrated by a sensitivity test that was performed with HeiPro and variable temperature inputs, results not shown). (6) The a priori NO₂ profile used in the retrievals may also contribute to the HeiPro bias towards Pandora-DS. Although a conservative surface VMR of ~3.5 ppbv was used for the a priori profile, it is difficult to know if the upper layers of the a priori profile are

conservative estimates, given the limited in situ profile information at this site. It is possible that the VMR values in the upper layers of the a priori profile are larger than the true values, which would contribute to the larger HeiPro bias relative to Pandora-DS.

Additionally, we investigated whether the $NO_2$ heterogeneity that we believe is contributing to the HeiPro bias to Pandora-DS can be supported by TROPOMI and GEM-MACH $NO_2$ fields at the measurement site. Overall, we were not able to utilize these datasets to show strong $NO_2$ heterogeneity between the two viewing geometries during the early morning hours

and during the winter months. This suggests that satellite measurements and model output may not capture small-scale $NO_2$ enhancements: TROPOMI provides a smoothed representation of the $NO_2$ field and lacks sensitivity to the boundary layer due to the use of low-resolution a priori profiles, while GEM-MACH may not accurately represent real-time $NO_2$ gradients and intensities. Pandora MAX-DOAS measurements may therefore provide a better tool for probing lower tropospheric $NO_2$ and its heterogeneity around a measurement site since the sensitivity and temporal coverage of satellites are limited, in situ

measurements are spatially limited, and the direct-Sun measurement viewing geometry is subject to the Sun's position and clear-sky conditions.

While HeiPro $NO_2$ partial columns are larger than Pandora-DS, GEM-MACH, and TROPOMI, HeiPro surface $NO_2$ agrees reasonably well with NAPS in situ, with a mean relative bias of $-9.7\% \pm 7.5\%$, and appears to underestimate GEM-

725 MACH, with a mean relative bias of $-37\% \pm 2.4\%$. This underestimation of GEM-MACH was consistent across all measurement hours of the day and all seasons, while HeiPro more closely matched the diurnal and seasonal trends of NAPS in situ surface $NO_2$. In comparing the HeiPro and GEM-MACH $NO_2$ profiles, HeiPro generally underestimates GEM-MACH in the 0–200 m layer with a mean relative bias of $-37\%$, and overestimates GEM-MACH in the 1.5–4 km layer with a significantly large mean relative bias > 1000%. These discrepancies as a function of altitude can probably be explained by the

730 GEM-MACH model inventories. On the other hand, the smoothed GEM-MACH profiles more closely match the HeiPro profiles across all seasons (mean relative bias from 0–200 m: $-6.1\%$; mean relative bias from 1.5–4 km: 2.6%) as the measurement limitations and vertical sensitivity are removed in the comparison. In summary, the three-year $NO_2$ profile dataset presented in this study provides information about the spatiotemporal vertical distribution of $NO_2$ at the Downsview measurement site and can be used to assess discrepancies between spaceborne and ground-based $NO_2$ measurements.

**Appendix A**

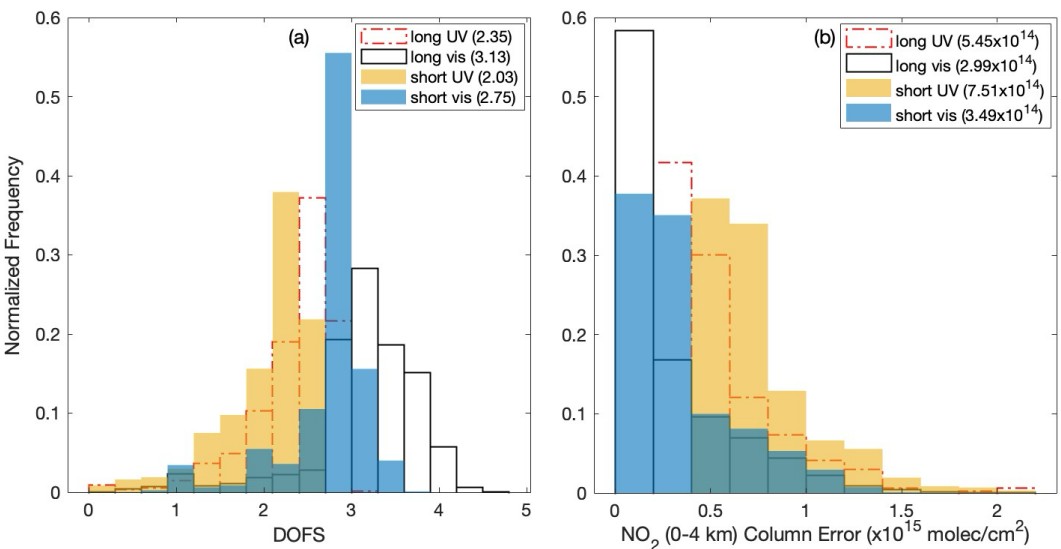

Figure A1: (a) Histogram of the normalized frequency of degrees of freedom for signal (DOFS) for the HeiPro

retrieved $NO_2$ profiles from each type of Pandora multi-axis scan, where the mean values are indicated in brackets.

(b) Same as (a) but for $NO_2$ partial column errors.

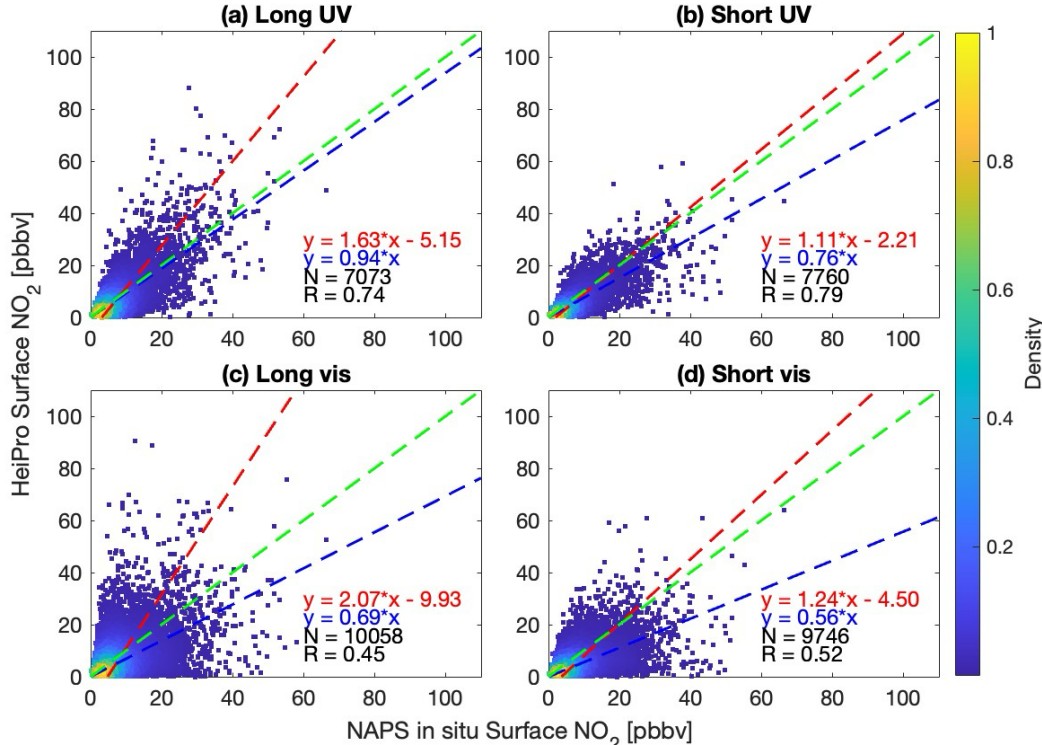

**Figure A2. Scatter plots for HeiPro surface NO₂ vs. NAPS in situ surface NO₂ VMRs for scan types and retrieval windows of (a) long UV, (b) short UV, (c) long vis, and (d) short vis. Only the long UV scans were incorporated in the results of this work. The dashed lines and color bar are as indicated in Fig. 3.**

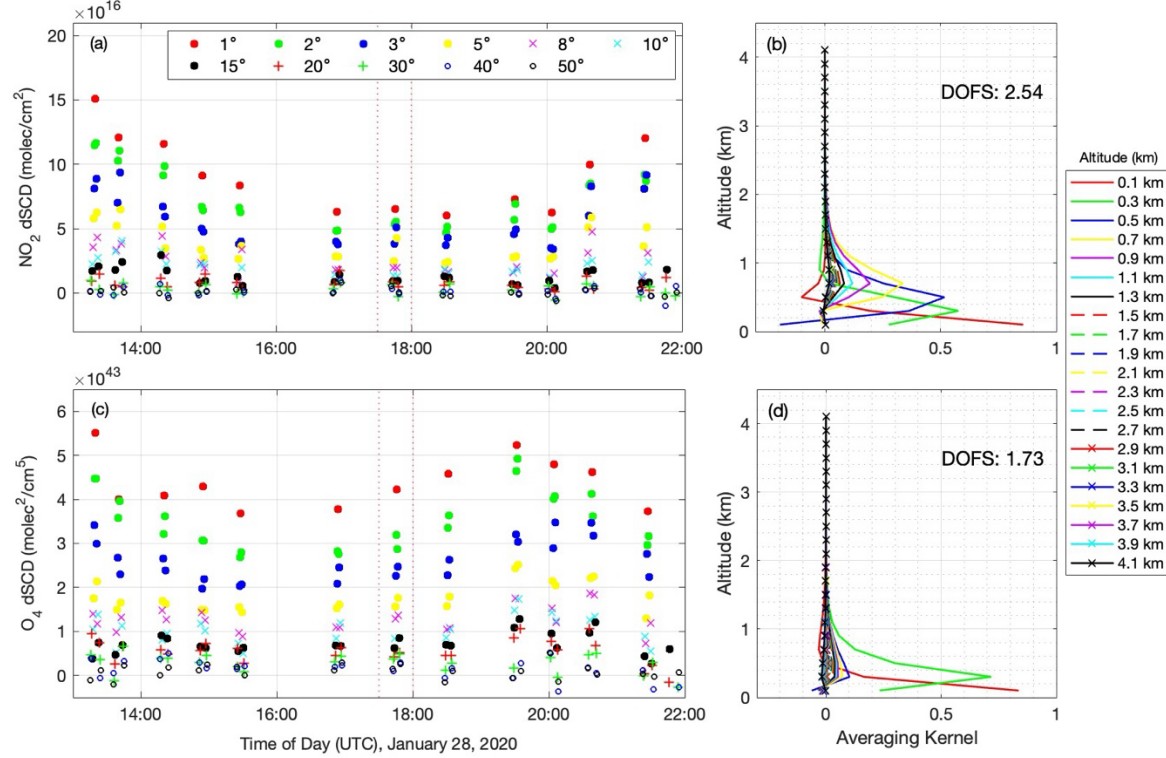

**Figure A3: MAX-DOAS dSCDs (UV, 338–370 nm) of (a) NO₂ and (c) O₄ retrieved on January 28, 2020. The right panels show the averaging kernels and respective DOFS for a single HeiPro profile retrieval of (b) NO₂ and (d) aerosol extinction. The dSCDs used in the NO₂ and aerosol extinction profile retrievals are indicated by the red dashed lines in panels (a) and (c), respectively.**

**Table A1. Multiplicative biases and mean relative biases (± uncertainties) of HeiPro towards NO₂ partial columns from Pandora-DS, TROPOMI, and GEM-MACH, for both the HeiPro long UV and long vis results.**

| NO₂ Partial Column Comparison | | HeiPro Scan Type | |
|---|---|---|---|
| | | long UV | long vis |
| **HeiPro vs. Pandora-DS partial columns** | Multiplicative Bias | $51\% \pm 0.8\%$ | $49\% \pm 0.9\%$ |
| | Mean Relative Bias | $61\% \pm 9.7\%$ | $61\% \pm 6.8\%$ |
| **HeiPro vs. TROPOMI** | Multiplicative Bias | $17\% \pm 4.0\%$ | $13\% \pm 4.6\%$ |
| | Mean Relative Bias | $37\% \pm 51\%$ | $40\% \pm 45\%$ |
| **HeiPro vs. GEM-MACH** | Multiplicative Bias | $12\% \pm 1.2\%$ | $13\% \pm 1.3\%$ |
| | Mean Relative Bias | $67\% \pm 7.1\%$ | $64\% \pm 2.4\%$ |

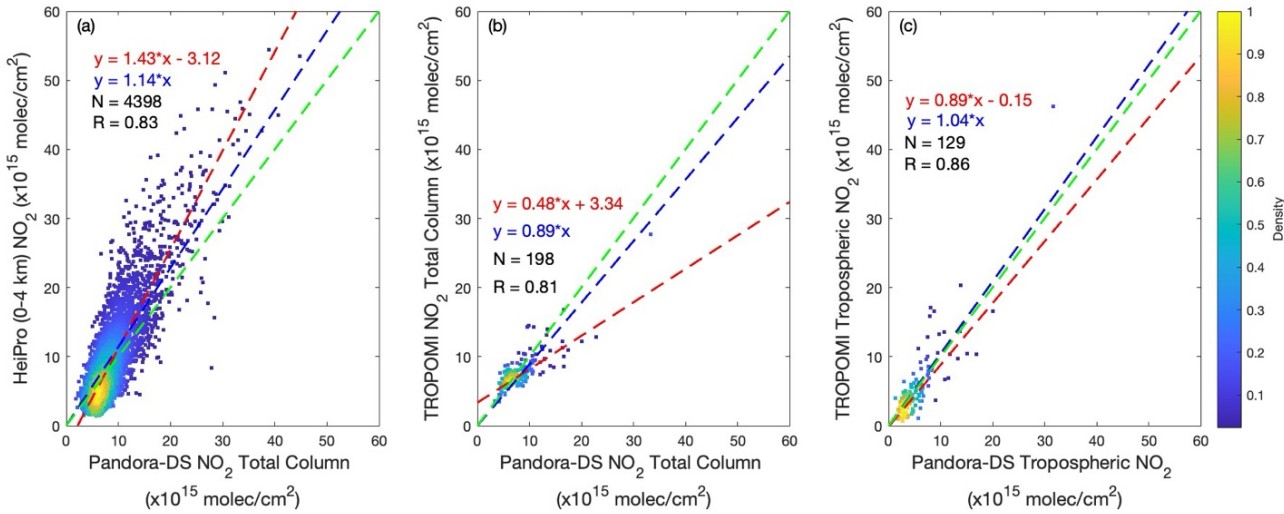

**Figure A4**. **Comparisons (2018–2020) of (a) HeiPro (0–4 km) NO₂ partial columns vs. Pandora-DS NO₂ total columns, (b) TROPOMI vs. Pandora-DS NO₂ total columns, and (c) TROPOMI vs. Pandora-DS tropospheric NO₂ columns. The dashed lines and color bar are as indicated in Fig. 3.**

In Figs. A5 and A6, various measurement lines of sight are depicted. Figure A5a–b displays the TROPOMI pixel-
770 averaged NO₂ field from 2018–2020 for summer and winter months, respectively. The MAX-DOAS azimuth viewing angle
(255°) and direct-Sun viewing angle (average during summer or winter time periods) are shown. A marker indicates the
horizontal extent of the multi-axis path length, i.e., the effective path length. For UV MAX-DOAS measurements, this value
is in the range of 5–10 km, and so a horizontal path length of 7.5 km is indicated in the figures (Ortega et al., 2015). During
both summer and winter, there does not appear to be an obvious difference in NO₂ between the Pandora MAX-DOAS azimuth
viewing angle and the direct-Sun viewing angle at this time of day. However, there are limitations to using the TROPOMI
NO₂ field to investigate the HeiPro bias to Pandora-DS, since the TROPOMI data itself underestimates HeiPro and may be
insufficient to address how NO₂ heterogeneity from local emissions contributes to the HeiPro overestimation (discussed in
Section 3.2). The single temporal datapoint from TROPOMI (1:30 LST) also prevents an investigation as to how
spatiotemporal NO₂ heterogeneity around Downsview is impacting the HeiPro overestimation. Alternatively, the GEM-
780 MACH dataset allows an exploration of how the NO₂ heterogeneity changes throughout the day and is discussed next.

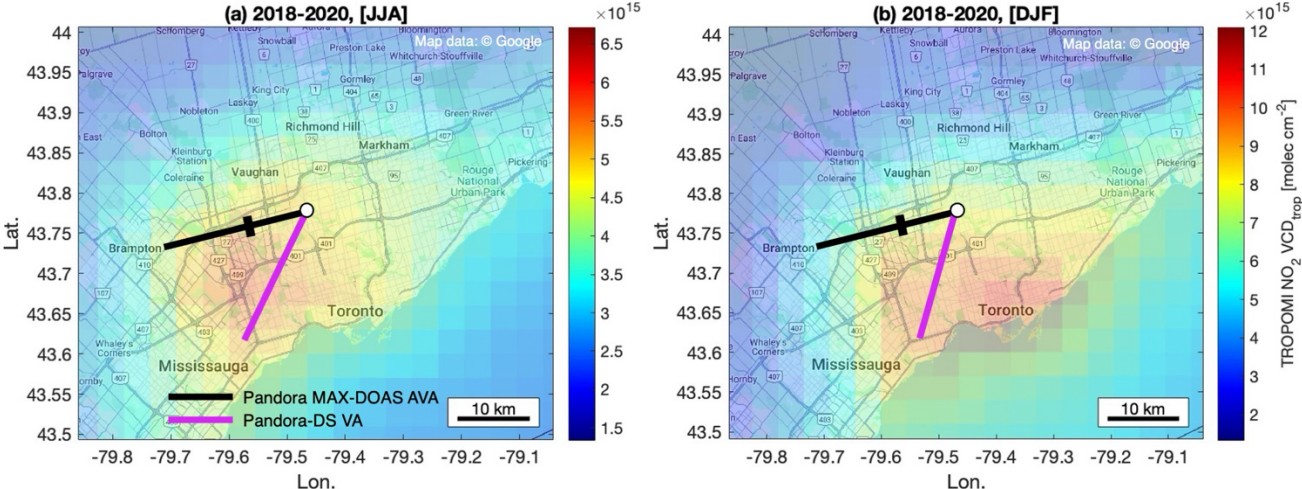

**Figure A5.** A map (© Google Maps) of the measurement site (Downsview, white circle) and surrounding Greater Toronto Area masked by the TROPOMI pixel-averaged tropospheric NO₂ field from 2018 – 2020 during (a) summer only and (b) winter only. The color bar indicates the NO₂ tropospheric column in molec cm⁻². The black line represents the Pandora instrument's multi-axis azimuth viewing angle (AVA) of 255° and the magenta line represents the mean direct-Sun viewing angle during the TROPOMI overpass time (13:30 LT). The average MAX-DOAS effective path length of 7.5 km is depicted by the black marker along the line of sight.

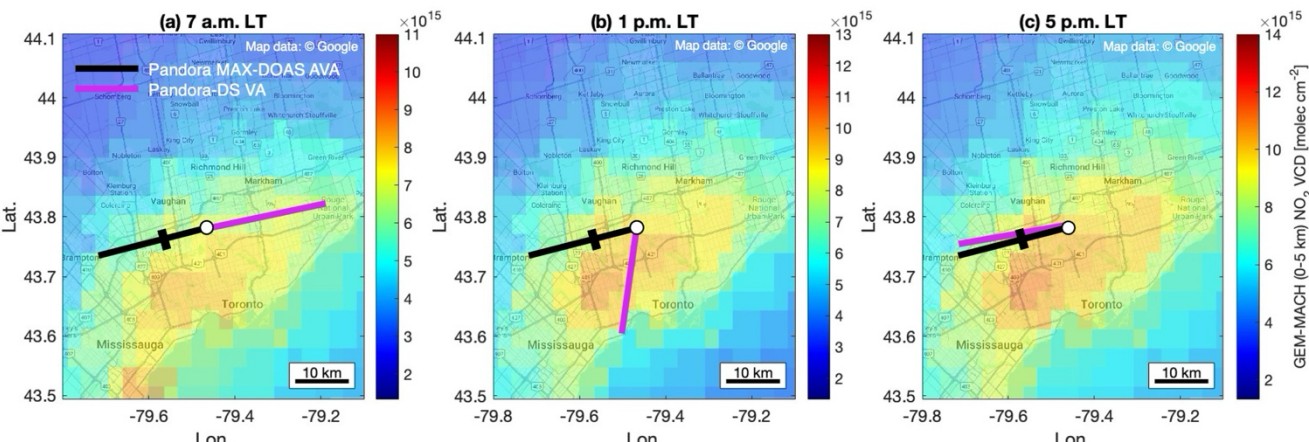

**Figure A6.** A map (© Google Maps) of the measurement site (Downsview, white circle) and surrounding Greater Toronto Area masked by the GEM-MACH averaged NO₂ field from 2018–2020 at (a) 7 a.m. LT, (b) 1 p.m. LT, and (c) 5 p.m. LT. The black line represents the Pandora instrument's multi-axis azimuth viewing angle (AVA) of 255° and the magenta line represents the mean Pandora direct-Sun viewing angle during the corresponding times. The average MAX-DOAS effective path length of 7.5 km is depicted by the black marker along the line of sight.

In Fig. A6a–c, the 2018–2020 mean $NO_2$ field from GEM-MACH is displayed for 7 a.m., 1 p.m., and 5 p.m. LT, respectively, with the Pandora-DS viewing angles shown for each time of day. From the 7 a.m. field, it is not evident that a spatially heterogeneous $NO_2$ field, combined with differences in viewing geometries, is contributing to the larger bias of HeiPro to Pandora-DS during the early morning hours, since the direct-Sun viewing angle at this time of day faces a similarly polluted region to the multi-axis viewing direction. Although Fig. 5 shows that the bias decreases throughout the day as the sampling directions approach one another, the difference in $NO_2$ levels between the two sampling directions in the GEM-MACH $NO_2$ field does not reflect that, possibly due to limitations in using model data to capture local enhancements in $NO_2$.

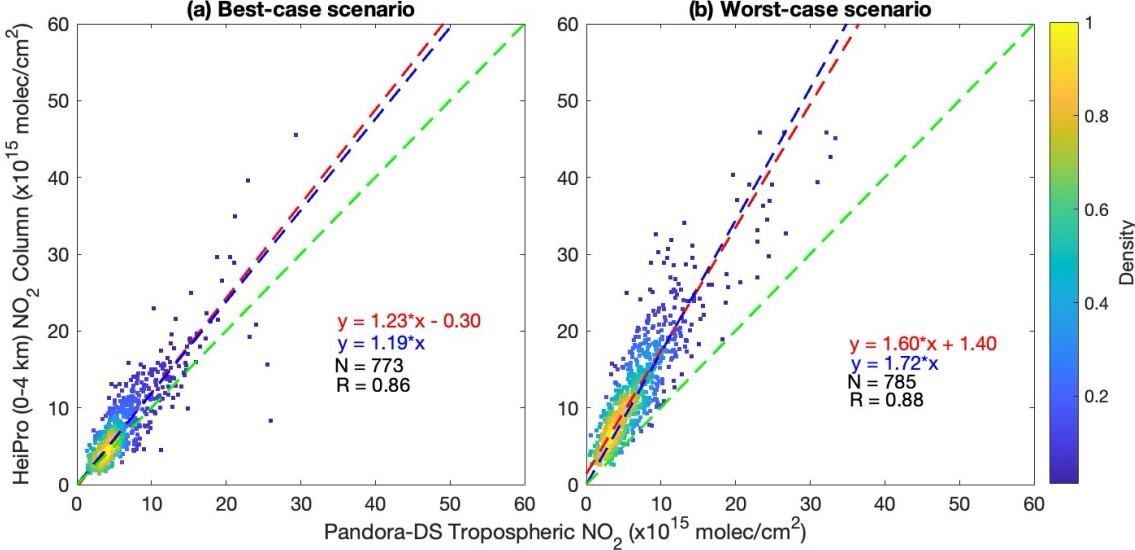

**Figure A7. Scatter plots for HeiPro (0–4 km) vs. Pandora-DS tropospheric $NO_2$ columns under (a) best-case scenario conditions (utilizing modified Pandora-DS tropospheric columns during summer months with measurements of SAA > 125°) and (b) worst-case scenario conditions (utilizing standard Pandora-DS tropospheric columns during (i) winter months and (ii) 6–7 a.m. across remaining seasons). The dashed lines and color bar are as indicated in Fig. 3.**

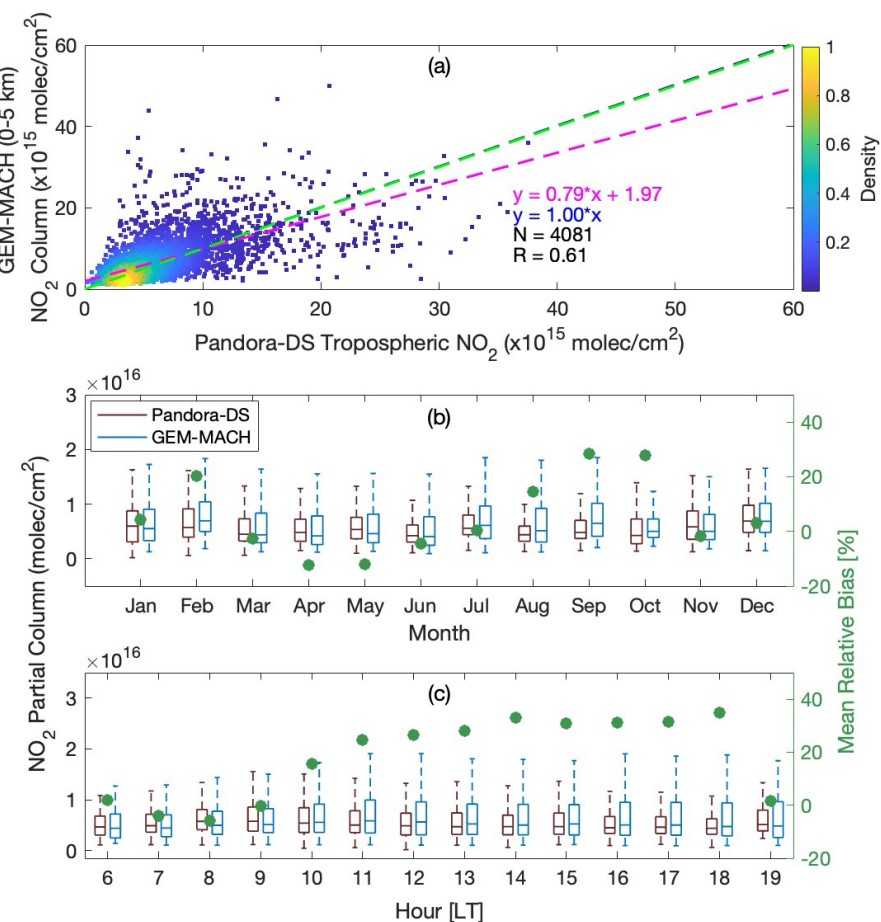

**Figure A8. (a) Scatter plot for the GEM-MACH (0–5 km) vs. Pandora-DS tropospheric NO₂. The dashed lines and color bar are as indicated in Fig. 3. (b) Monthly and (c) hourly box-and-whisker plots of Pandora-DS tropospheric (brown) and GEM-MACH (0–5 km, blue) NO₂ columns as well as the mean relative bias between the two (green circles). Box-and-whisker values are as defined in Fig. 4.**

## Data availability

Pandora data (L1 multi-axis spectra and L2 direct-Sun $NO_2$ total columns) are available from the Pandonia network (https://data.pandonia-global-network.org/Downsview/Pandora103s1/; Pandonia Global Network, 2024). OMI $NO_2$ SPv3.1 data are available from https://doi.org/10.5067/Aura/OMI/DATA2017 (Krotkov et al., 2019). TROPOMI L2 $NO_2$ data are available on the Copernicus Open Access Hub at https://scihub.copernicus.eu (ESA, 2024). The NAPS in situ data can be downloaded from https://data-donnees.az.ec.gc.ca/data/air/monitor/national-air-pollution-surveillance-naps-program/, last accessed on 10 September 2024. The HeiPro data ($O_4$ and $NO_2$ dSCDs that went into the retrievals as well as the subsequent profile retrievals) are made available at https://borealisdata.ca/dataset.xhtml?persistentId=doi:10.5683/SP3/J8PDHW.

## Author contribution

RA, XZ, KB, and KS conceived this project. RA was responsible for conducting the data analysis, creating all the figures and plots, and drafting the paper. XZ, KB, and KS provided significant inputs throughout the study, and XZ, DG, VF, UF, KB, and KS provided significant edits to the paper. JD, VF, XZ, and SCL operated and maintained the Pandora103 instrument. DG and AL provided the GEM-MACH data products. UF developed the HeiPro algorithm. YS provided the NAPS in situ data. AC managed the Pandonia network and provided vital technical support for the Downsview site and the ensuing data analysis. All coauthors provided feedback on the paper.

## Competing interests

At least one of the authors is a member of the editorial board of *Atmospheric Measurement Techniques*.

## Acknowledgments

Ramina Alwarda was supported by the ECCC Research Affiliate Program and funding from the University of Toronto Faculty of Arts & Science and would like to express her gratitude for this support. We would like to thank the ECCC Grants & Contributions Program (contribution agreement no. GCXE22S069) for their financial support, which also made this research possible. We acknowledge the NASA Earth Science Division for providing the OMI NO2 SPv3.0 data as well as the Netherlands Space Office (NSO) and the European Space Agency (ESA) for providing the Sentinel-5 Precursor TROPOMI Level 2 product and appreciate the efforts of all those involved in the collection, processing, and distribution of the satellite data. We are grateful to the Pandonia Global Network team and collaborators for their invaluable contributions and ongoing support in the operation and maintenance of the Pandora spectrometer network. The PGN is a bilateral project supported with funding from NASA and ESA. We also thank Vitali Fioletov and staff for their effort in establishing and maintaining Pandora103 at the Downsview site. We acknowledge the European Centre for Medium-Range Weather Forecasts (ECMWF) for providing the ERA5 reanalysis data used in this study, which is publicly available through the Copernicus Climate Change Service (C3S). The authors would like to thank the anonymous reviewers for providing constructive feedback, which helped improve the quality of this manuscript.

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
