# Peer review of "Retrieval of NO2 profiles from three years of Pandora MAX-DOAS measurements in Toronto, Canada"

_Atmospheric Measurement Techniques, 2024_

## Author Comment (AC1)

**Response to AMT Reviewer #1**

The authors present a novel study using Pandora MAX-DOAS measurements and the HeiPro retrieval algorithm to produce a three-year dataset of NO2 profiles and partial columns in Toronto, Canada. While the paper provides valuable insights into the spatial and temporal distribution of NO2. However, several areas require clarification, reorganization, and deeper analysis to strengthen the validity and comprehensibility of the study. My detailed comments are below.

We thank the reviewer for their helpful comments, which have helped to improve the manuscript. Our point-by-point responses are provided below in blue font, with new text added to the manuscript given in *red font and italics*. Line numbers refer to the new clean version of the manuscript.

**General Comments**

**Stratospheric-Tropospheric Separation (STS) Method**
- The authors use a complicated approach involving a box model and OMI observations to separate stratospheric and tropospheric columns for Pandora direct-sun (DS) measurements. The STS method is questionable due to multiple layers of assumptions and models, creating uncertainties.
  - Why not employ stratospheric-tropospheric column ratios from established models, such as CAMS, TM5, or GEM-MACH, to simplify and enhance the accuracy?
    - The STS method used in this work follows the approach of several previous studies (e.g., Zhao et al., 2019, Choi et al., 2020, Zhao et al., 2022) , while the OMI stratospheric NO$_2$ data product (v3) has been shown to have good agreement with other satellite and ground-based FTIR measurements (Krotkov et al., 2017). Additionally, TROPOMI vs. Pandora-DS tropospheric NO$_2$ show good agreement (discussed further in the next bullet point), providing an independent verification of the STS method used. Models such as CMAS and TM5 are not independent data sources, if we wish to compare the results with satellite instruments (such as TROPOMI). For example, TM5 has already been used in TROPOMI STS algorithm, CAMS assimilated TROPOMI data, etc. GEM-MACH is a good option, but the operational version involved in current work only has tropospheric column. However, we fully agree with the referee that we could explore some more advanced models for the STS in the future.
    - Additionally, the stratospheric portion removed from the Pandora-DS total columns is 34% $\pm$ 2.8%. For reference and comparison, we also calculated the stratospheric-to-total column portion for TROPOMI measurements and found this value to be 44% $\pm$ 9.7%. Please note that the former value represents data throughout the Pandora-DS measurement day while the latter represents data at a single measurement time per day. The following

text has been revised in the manuscript, to include the stratospheric percentage, on lines 215–216:

*Due to the diurnal variation of $NO_2$, and the satellite's overpass time of 13:30 local time (LT), a photochemical box model (Pratmo, discussed further in Section 2.3.1) was used to calculate stratospheric $NO_2$ at various Pandora measurement times throughout the day. The stratospheric portion that was removed accounted for 34% $\pm$ 2.8% of the Pandora-DS $NO_2$ total columns.*

o A direct comparison of Pandora-DS total columns with TROPOMI total columns would provide additional insights into discrepancies between ground-based and satellite observations.

- A plot of TROPOMI vs. Pandora-DS $NO_2$ total columns has now been added in the manuscript in the Appendix (Figure A4) and is copied below for reference. For total column comparisons, the mean relative bias of TROPOMI to Pandora-DS is 1.53% $\pm$ 21.2% while the zero-intercept slope is 0.89 $\pm$ 0.02. For tropospheric column comparisons, the mean relative bias of TROPOMI to Pandora-DS is $-0.85\%$ $\pm$ 34% while the zero-intercept slope is 1.04 $\pm$ 0.03. At this measurement site, ground-based direct-Sun and satellite observations of $NO_2$ agree reasonably well. The agreement between the tropospheric columns provides an indication that the stratospheric-tropospheric separation method we are using is reasonable.

- The following text has been added to the manuscript on lines 461–475:

*For reference, HeiPro (0–4 km) vs. Pandora-DS total columns are compared in panel (a) of Fig. A4, which shows that HeiPro partial columns exhibit a positive multiplicative bias of 16% $\pm$ 0.7% and a mean relative bias of 6.1% $\pm$ 4.8%. Not surprisingly, there is better agreement here as compared to Fig. 3a (i.e., HeiPro vs. Pandora-DS tropospheric $NO_2$) since the Pandora-DS total columns are larger. The TROPOMI vs. Pandora-DS $NO_2$ total and tropospheric column comparisons are shown in Fig. A4b–c, respectively. Pandora-DS and TROPOMI show good agreement with one another for both total column (multiplicative bias: $-12\%$ $\pm$ 1.9%; mean relative bias: 0.1% $\pm$ 21%) and tropospheric $NO_2$ (multiplicative bias: 4.4% $\pm$ 3.5%; mean relative bias: $-0.9\%$ $\pm$ 34%). Note that the large uncertainties are due to the relatively large TROPOMI total column and tropospheric $NO_2$ errors. Additionally, the tropospheric $NO_2$ agreement in panel (c) provides more confidence in the stratospheric-tropospheric separation method that was used in the study (i.e., Pratmo-OMI data).*

[Figure]

**Figure A4**. Comparisons (2018–2020) of (a) HeiPro (0–4 km) NO₂ partial columns vs. Pandora-DS NO₂ total columns, (b) TROPOMI vs. Pandora-DS NO₂ total columns, and (c) TROPOMI vs. Pandora-DS tropospheric NO₂ columns. The dashed lines and color bar are as indicated in Fig. 3. For panels (b) and (c), the coincidence criteria are 10 km and 10 minutes.

**MAX-DOAS Retrieval and Atmospheric Profile**
- The description of the MAX-DOAS profile retrieval lacks essential details:
  - Are the ERA5 atmospheric profiles daily averages or spatio-temporally interpolated to the measurement times?
    - The ERA5 atmospheric profiles are daily averages obtained from the ERA5 grid box nearest to the measurement site. The atmospheric profiles are calculated from the average of daytime (11:00, 14:00, 17:00, 20:00, 23:00 UTC) temperature and pressure profiles. This is discussed in Section 2.4.2 (lines 367–369), which describes the ERA5 data used in the study, but the following text has been added in Section 2.1.3 (lines 280–281) for clarification:
      *Additional a priori inputs to HeiPro were daily pressure and temperature profiles from ERA5 reanalysis data at the grid box nearest to the measurement site (discussed further in Section 2.4.2).*
  - What assumptions are made about NO2 above the retrieval height? Are these values based on standard atmospheric profiles or other sources?
    - The retrieval height is 0–4 km. For the HeiPro datasets presented here, NO₂ above the retrieval height is not considered. Therefore, the HeiPro profiles that are integrated to produce columns represent partial columns from 0–4 km. This has been stated in the manuscript text and all figures where necessary.

**Data Consistency and Filtering**
- The differences in coinciding data points in Figure 3(a) (direct sun) and Figure 3(c) (model comparisons) require clarification:
  - If Pandora DS and MAX-DOAS data originate from the same source, why is there a discrepancy in the data points?

- The reason for the discrepancy in the number of coincident data points between Fig. 3a and Fig. 3c is due to data gaps in the OMI $NO_2$ stratospheric column data used in the calculation of the Pandora-DS tropospheric columns. While MAX-DOAS and direct-Sun have slightly different measurement schedules, they do have a similar measurement frequency as they are from the same instrument. However, the OMI stratospheric $NO_2$ data (which is used for the Pandora-DS tropospheric data) has gaps with missing days and hours of the day. Therefore, this shortens the number of coincident data points between Pandora-DS and the MAX-DOAS HeiPro data, while the MAX-DOAS HeiPro and GEM-MACH data do not have this temporal limitation. We have modified panels (a) and (c) in Fig. 3 to only include coincident data among HeiPro, Pandora-DS, and GEM-MACH, and so they have the same number of data points ($N$=3284). We have also adjusted any reference in the text to the biases of HeiPro to GEM-MACH partial columns (they changed very slightly due to the new number of data points). The modified Fig. 3 is copied below for reference:

[Figure]

○ Did the authors apply data filtering, such as cloud filtering, before using Pandora data for MAX-DOAS retrievals?
- Yes, Pandora $O_4$ and $NO_2$ dSCDs with fitting residual RMS values ≥ 0.003 for the dSCD retrieval were filtered out (only 5% of the data, mostly for SZA > 80°). This filter was selected to improve the quality of the fits and discard noisier fits from the analysis. This has now been clarified in Section 2.1 of the manuscript (lines 268–272):

  *Following the QDOAS analysis, $O_4$ and $NO_2$ dSCDs with fitting residual root-mean-square (RMS) values ≥ 3×$10^{-3}$ were discarded (only 5% of the data, mostly at SZA > 80°). This filter was selected to improve the quality of the fits and discard noisier fits from the analysis.*
- Cloud filtering was not used.

- o Similarly, in Figure 4, the differences between MAX-DOAS (HeiPro) results in panels (a) vs. (c) and (b) vs. (d) need to be explained.
    - ▪ The differences between HeiPro in panel 4a vs. HeiPro in panel 4c is due to the same reason listed under the bullet point above (i.e., that the OMI stratospheric NO₂ data limits the number of coinciding data points with MAX-DOAS and Pandora-DS). Figure 4 has been modified to plot data that is coincident among HeiPro, Pandora-DS, and GEM-MACH. Therefore, the HeiPro results in panels (a) vs. (c) and (b) vs. (d) are now identical. The modified Fig. 4 is copied below for reference.

[Figure]

**Organization and Logical Flow**
- Figure 1 is currently located in the introduction but would fit better in Section 2, "Instrument Description," to align with the discussion of the Pandora instrument and measurement conditions.
    - o We agree with the suggestion that Figure 1 is better suited for Section 2. Figure 1 (now Figure 2) and corresponding text have been moved to Section 2.
- Section 2 should follow a logical sequence: instrument description, retrieval algorithm, and then the models/data used for comparison and validation. This reorganization would enhance the clarity of the methods section.
    - o Section 2 has been rearranged so that Section 2.1 focuses solely on Pandora data. The Pandora is introduced and the three subsections that follow go into detail regarding the direct-Sun, dSCD, and profiling retrieval algorithms. Section 2.2 then discusses the TROPOMI instrument and measurements. Section 2.3 discusses the in situ instrument and measurements. Section 2.4 and the three subsections that follow discuss the model data (ERA5, PRATMO, and GEM-MACH). We believe that

this reorganization is clearer and more concise, with each dataset described along with, or shortly after, its respective instrument description.

**Wavelength Range and Spectral Retrieval**
- The wavelength ranges for direct-sun and MAX-DOAS spectral fits are unclear:
  - Did the authors use the same wavelength range for both retrievals? If not, provide justification for the differences.
    - The wavelength ranges for direct-Sun and MAX-DOAS retrievals are different. The wavelength range for Pandora direct-Sun is 400–440 nm, as per the PGN's direct-Sun $NO_2$ retrieval algorithm. This retrieval window has been verified, has a well-known performance, and is currently the official data product from PGN (e.g., Zhao et al., 2019). The wavelength range for MAX-DOAS is 338–370 nm, which is based on the MAX-DOAS UV retrieval guidelines for $NO_2$ outlined in Kreher et al. (2020) and is a recommended protocol by the Network for the Detection of Atmospheric Composition Change (NDACC) UV-visible working group. The wavelength ranges were previously stated in the manuscript in Section 2 where the retrieval algorithms are discussed and in the Conclusions section. Although the retrieval windows differ, it is worth noting that even when comparing the MAX-DOAS long vis HeiPro results to Pandora-DS, the results are very similar to the long UV HeiPro results (see table below), indicating that the differences introduced by the choice of fitting window is small. The following text has been added to the manuscript on lines 422–431:
    *Although the direct-Sun and MAX-DOAS retrieval wavelengths are different due to the varying standard protocols for each, it is worthwhile to note that the HeiPro long vis versus long UV $NO_2$ partial column comparisons showed remarkable agreement with one another, with a zero-intercept slope of $0.97 \pm 0.004$ and mean relative bias of $0.7\% \pm 5.9\%$. We therefore do not expect the choice of retrieval window to significantly impact the HeiPro long UV partial column comparisons to Pandora-DS (see Table A1 for the HeiPro long vis $NO_2$ partial column comparisons).*
  - Details of the spectral retrieval process should be included.
    - We have added more detail on the direct-Sun spectral retrieval process. The following text has been added to the manuscript on lines 201–208:
    *The standard Pandora-DS total column $NO_2$ data product is obtained using Total Optical Absorption Spectroscopy (TOAS), as implemented by PGN's BlickP software (Cede, 2019). Direct-Sun spectra in the 400–440 nm range are fitted with cross-sections of $NO_2$ (at an effective temperature of 254.5 K, Vandaele et al., 1998), $O_3$ (at an effective temperature of 255 K, Brion et al., 1993, 1998; Daumont et al., 1992), and a fourth-order polynomial to produce SCDs of $NO_2$ with a clear-sky precision of $2.7 \times 10^{14}$ molec $cm^{-2}$ (Herman et al., 2009). A synthetic reference spectrum is used in the analysis and is obtained by taking an average of several measured spectra which are corrected for their total optical depth. Following this, $NO_2$ SCDs are*

> *converted to vertical column densities (VCDs or total columns) using geometric AMFs. The Pandora-DS NO$_2$ VCD has an absolute accuracy of 1.3×10$^{15}$ molec cm$^{-2}$ (Herman et al., 2009).*

- ▪ We also now include more detail on the MAX-DOAS spectral retrieval process. The following text has been added to the manuscript on lines 260–268:

  *O$_4$ and NO$_2$ dSCDs were retrieved in both the ultraviolet (UV, 338–370 nm, only measurements with the UV band pass filters) and visible (vis, 425–490 nm) windows. Differential cross-sections of NO$_2$ at 294 and 220 K for both windows (Vandaele et al., 1998), O$_4$ at 293 K for both windows (Thalman and Volkamer, 2013), O$_3$ at 223 and 243 K for UV and 223 K for vis (Serdyuchenko et al., 2014), BrO at 223 K for UV only (Fleischmann et al., 2004), HCHO at 297 K for UV only (Meller and Moortgat, 2000), and H$_2$O for vis only (Rothman et al., 2010) were convolved using the instrument slit function and the nominal wavelength calibration file from PGN. A fifth-degree polynomial, linear offset, and first-order shift and stretch were used in both windows (Kreher et al., 2020).*

- ○ Line 217-218 mentions NO2 retrieval in both UV and VIS bands—did the authors compare results from these bands? Specify which band was used for comparisons with direct sun, satellite, and model data.

  - ▪ The manuscript had previously stated that only the UV results were used for all comparisons in the manuscript (line 308). We did not initially include comparisons of HeiPro long UV vs. long vis in the manuscript, but the results are shown in (i) the scatter plot below which shows a zero-intercept slope of 0.97 ± 0.004 and mean relative bias of 0.67% ± 5.9%, and (ii) the table below which shows that the HeiPro long UV and long vis biases towards the partial columns are similar (mostly the same within uncertainties). The table has been added to the manuscript in the Appendix (Table A1).

[Figure]

**Table A1. Multiplicative biases and mean relative biases ($\pm$ uncertainties) of HeiPro towards NO$_2$ partial columns from Pandora-DS, TROPOMI, and GEM-MACH, for both the HeiPro long UV and long vis results.**

| NO$_2$ Partial Column Comparison | | HeiPro Scan Type | |
|---|---|---|---|
| | | long UV | long vis |
| HeiPro vs. Pandora-DS partial columns | Multiplicative Bias | 51% $\pm$ 0.8% | 49% $\pm$ 0.9% |
| | Mean Relative Bias | 61% $\pm$ 9.7% | 61% $\pm$ 6.8% |
| HeiPro vs. TROPOMI | Multiplicative Bias | 17% $\pm$ 4.0% | 13% $\pm$ 4.6% |
| | Mean Relative Bias | 37% $\pm$ 51% | 40% $\pm$ 45% |
| HeiPro vs. GEM-MACH | Multiplicative Bias | 12% $\pm$ 1.2% | 13% $\pm$ 1.3% |
| | Mean Relative Bias | 67% $\pm$ 7.1% | 64% $\pm$ 2.4% |

**Quality Filtering and Averaging Kernels**

- Section 2.2.2 discusses filtering criteria (e.g., DOFS < 1), but the rationale behind these criteria is not well explained:
    - Provide theoretical or empirical justification for the chosen threshold.
        - Thank-you for flagging this. A DOFS threshold of 1 is a conservative value and indicates that there is one independent piece of information from the retrieval. It is used, for example, in a study by Vlemmix et al. (2015) for quality control of aerosol, formaldehyde, and NO$_2$ profile retrievals. The plot below shows in panel (a) the profiles that meet the DOFS threshold and in panel (b) the profiles that do not. As shown in panel (b), the mean of the NO$_2$ profiles with DOFS < 1 (solid black line) more closely resembles the a priori (dashed black line) when compared to the mean of the NO$_2$ profiles that meet the threshold in panel (a). Because profiles with DOFS < 1 largely resemble the a priori profile, they are discarded as they contain limited information from the measurements. We modified the manuscript on lines 327–329 to now say:

            *Lastly, retrievals for which both the NO$_2$ profiles and aerosol extinction profiles had DOFS < 1 were excluded from the analysis for quality control purposes (e.g., Vlemmix et al., 2015) and represented 19% of the dataset. Such retrievals contain limited information from the measurements and are more influenced by the a priori profile.*

[Figure]

- o What is the typical DOFS value in the retrieval? Including an averaging kernel plot would help visualize the retrieval sensitivity.
  - ▪ For the NO$_2$ UV retrievals, the mean DOFS value is 2.35 and the median DOFS value is 2.48. For the AOD UV retrieval, the mean DOFS value is 1.61 and the median DOFS value is 1.78. Histograms of the DOFS for the various NO$_2$ retrievals are already shown in Figure A1(a), but the mean values of the long UV retrievals of AOD and NO2 before and after filtering are now stated in the manuscript (lines 329–331):

    *The mean DOFS values before and after filtering are 1.61 $\pm$ 0.68 and 1.88 $\pm$ 0.42 for aerosol extinction, respectively, and 2.35 $\pm$ 0.49 and 2.39 $\pm$ 0.40 for NO$_2$, respectively.*
  - ▪ Thanks for this suggestion. Averaging kernel plots for O$_4$ and NO$_2$, together with timeseries of O$_4$ and NO$_2$ dSCDs, have been added to the Appendix (Figure A3). It is also displayed below for reference:

[Figure]

**Figure A3: MAX-DOAS dSCDs (UV, 338–370 nm) of (a) NO₂ and (c) O₄ retrieved on January 28, 2020. The right panels show the averaging kernels and respective DOFS for a single HeiPro profile retrieval of (b) NO₂ and (d) aerosol extinction. The dSCDs used in the NO2 and aerosol extinction profile retrievals are indicated by the red dashed lines in panels (a) and (c), respectively.**

  o Demonstrating the impact of data filtering on the results would also improve transparency.

  The profiles that were filtered out (with AOD and NO₂ DOFS < 1) represent 19% of the dataset, with the majority of this 19% removed due to the AOD DOFS being below 1. This mostly does not have a large impact on the data (for example, partial column comparisons of HeiPro to Pandora-DS and TROPOMI remain essentially unchanged before and after filtering, but the comparisons to GEM-MACH do change). We have now included a statement that the DOFS filter excludes 19% of the profile retrievals as noted above (lines 327–329).

**Comparison of Data Sources**
- In the results section, the biases between HeiPro, Pandora-DS, TROPOMI, and GEM-MACH are analyzed, but the spatial, temporal, and observational characteristics of these datasets are not sufficiently discussed:
  o Summarize the resolutions, error characteristics, and limitations of each dataset in a table for clarity.
    ▪ We have added Table 1 in the manuscript, which includes the temporal, spatial resolutions, and errors/uncertainties of each dataset and is copied below for reference. We also add the following on lines 178–179:

*Table 1 provides a summary of the datasets used in the study as well as some characteristics of the datasets such as resolutions and uncertainties.*

**Table 1. Overview of the key attributes of the NO₂ datasets used in this study.**

| Dataset | Temporal resolution | Horizontal resolution | Errors/uncertainties |
|---|---|---|---|
| **Pandora-DS total column NO₂** | 90 s | < 4 km (for SZA < 50°) 4–17 km (for SZA 50°–80°) (Herman et al., 2009) | $1.3 \times 10^{15}$ molec/cm$^2$ (Herman et al., 2009, 2018) |
| **OMI stratospheric NO₂** | 1 day | 13 km × 24 km (Krotkov et al., 2017) | $2 \times 10^{14}$ molec/cm$^2$ (Krotkov et al., 2017) |
| **HeiPro** | 22 minutes | 5–10 km (effective pathlength, Ortega et al., 2015) | $4.4 \times 10^{14}$ molec/cm$^2$ (this work) |
| **TROPOMI tropospheric NO₂** | 1 day | 7 km × 3.5 km (5.5 km × 3.5 km since Aug. 2019) | $8.5 \times 10^{14}$ molec/cm$^2$ (Eskes and Eichmann, 2019) |
| **GEM-MACH** | 1 hour | 10 km × 10 km | N/A |
| **In situ NO₂** | 1 minute | Point measurement | 0.4 ppbv (Thermo Scientific, 2015) |

- o Quantify the contributions of individual factors (e.g., PBL height, SAA, and seasonality) to observed biases rather than relying solely on trend descriptions.
  - ▪ Contributions of individual factors to the bias of MAX-DOAS (HeiPro) to Pandora-DS have been quantified. Since the contributions of these individual factors varied throughout the day, we report ranges and maximum values rather than a single contribution percentage. For example, we stated the following in the Conclusions section of the manuscript on lines 900–902: "The PBL height, combined with the missing 0–15 m partial column in the Pandora-DS measurements, contributed a maximum of 8.4% of the multiplicative bias and 18% of the mean relative bias in the morning hours, with these values declining to < 5% in the evening hours". This was similarly done for the solar azimuth angle contribution and seasonal contribution. In summary, the PBL, solar azimuth angle, and seasonality contributed up to 8%, 27%, and 39% of the multiplicative bias, respectively, and 18%, 52%, and 85% of the mean relative bias, respectively.
  - ▪ For HeiPro comparisons to TROPOMI and GEM-MACH, quantifying the contributions of individual factors was not feasible:
    - ▪ Since the SAA does not affect the GEM-MACH data, it was not analyzed as a factor contributing to the bias. Additionally, for comparisons of HeiPro and GEM-MACH partial columns, effects of PBL height were not analyzed (whereas for HeiPro vs. Pandora-DS

comparisons, they were due to the missing 15 m in the Pandora-DS measurements).

- Factors such as PBL height and SAA could not be quantified for HeiPro vs. TROPOMI due to the low temporal frequency of measurements per day, which limited the PBL heights and SAA ranges sampled. Similarly, seasonality could not be assessed for HeiPro vs. TROPOMI due to the low number of coincident measurements when separated by season.

**Seasonal and Diurnal Trends**

- Provide more detailed explanations for observed trends:
  - How do lower PBL heights in the early morning contribute to NO2 accumulation? Why are concentrations higher in winter? Correlate these trends with emission sources and meteorological conditions at the observation site.
    - The following text has been added to the manuscript on lines 481–484:
      *This difference in detection may further be amplified by shallower PBL heights during winter months and morning hours when the PBL height is smaller due to lower surface temperatures and less boundary layer dynamics. This leads to less vertical mixing of pollutants, with $NO_2$ accumulating near the surface (e.g., Lin and McElroy, 2010; Chan et al., 2018; Schreier et al., 2019) where it is not captured by the Pandora-DS measurements in the first 15 m.*
    - We now mention in the manuscript that $NO_2$ concentrations are higher in the winter due to (i) reduced sunlight and increased lifetime of $NO_2$, and (ii) increased heat-associated emissions during the winter months (lines 486–489):
      *Lastly, while $NO_2$ increases during the wintertime due to greater anthropogenic emissions from heat sources (e.g., Meng et al., 2018) and increased lifetimes due to decreased solar radiation, it is possible that increased emissions can contribute to the bias if there are more $NO_2$ emissions coming from the multi-axis azimuth viewing direction of 255° compared to the various direct-Sun viewing angles.*
      - We do not have $NO_2$ emissions data for the measurement site and surrounding region, but potential sources can be the local residential homes. Therefore, we only speculate that wintertime emissions may be higher in the direction of the multi-axis scans (255°).

**Extrapolation and Vertical Profiles**

- Discuss the limitations of linear extrapolation methods used in vertical NO2 profile retrievals. Explore non-linear methods for more accurate surface concentration estimates.
  - We agree that non-linear extrapolation methods may provide more accurate surface concentration estimates, and we will explore this for future studies when

performing such extrapolations. The linear extrapolation we used may be underestimating the NO$_2$ gradient from 100 m to 0 m. On average, the linear extrapolation used in this work increases the NO$_2$ by 1 ppbv from 100 m to 0 m. Without NO$_2$ profile information at the measurement site, it is difficult to know what a realistic increase would be from 100 m to 0 m, but we will explore this in the future. The following text has been added to the manuscript on lines 295–296 to address the limitations of the linear extrapolation:

*The linear extrapolation method used in this study produces NO$_2$ surface values that are, on average, 1 ppbv larger than the NO$_2$ value at 100 m. Therefore, this extrapolation method may underestimate the HeiPro surface NO$_2$ values that we report.*

- Analyze the changes in GEM-MACH profiles before and after smoothing across different height ranges, identifying layers with the most significant changes.
    - In the manuscript, we now state the following on lines 837–846:

    *Across all seasons, the HeiPro NO$_2$ median profiles from 0–200 m underestimate the unsmoothed GEM-MACH median values, while from 1.5–4 km, the HeiPro median profiles then overestimate the unsmoothed GEM-MACH median values. For the 0–200 m layer, the mean relative bias of HeiPro towards GEM-MACH decreases from −37% (unsmoothed) to −6.1% (smoothed). Note that these biases are representative of the integrated 0–200 m layer and may differ slightly from the surface values reported in Table 2. The most significant changes occur in the layer from 1.5–4 km, where the HeiPro bias towards GEM-MACH decreases from > 1000% (unsmoothed) to 2.6% (smoothed). The HeiPro surface underestimation and free tropospheric overestimation of the unsmoothed GEM-MACH profiles can probably be explained by the NO$_2$ inventories used in the GEM-MACH model, which, respectively, (i) utilize older inventories that do not account for reduced emissions over the years, and (ii) do not account for free tropospheric NO$_2$ sources while the a priori NO$_2$ profile contains free tropospheric NO$_2$.*

**Minor Comments**

**Technical Issues**
1. **Line 225**: The term "DOF" is inconsistently introduced. Ensure its full form and abbreviation are aligned throughout the text.
    - The full form (degrees of freedom for signal) and abbreviation (DOFS) are defined on lines 314–315. We have checked that DOFS is subsequently used consistently within the text.
2. Standardize terminology for "Pandora direct-Sun" and "Pandora-DS" to avoid confusion.
    - The full form (Pandora direct-Sun) and abbreviation (Pandora-DS) are defined on line 120. We have checked that Pandora-DS is subsequently used consistently within the text.

**Figures and Tables**
- Figures 3 and 4: Clarify the differences in datasets and ensure consistent labeling.

- - o Differences in: (i) the number of points in Figs. 3a versus 3c, and (ii) the box-and-whisker plots in Fig. 4 are as discussed above. The y-axis label in Fig. 3a has been changed to "HeiPro (0–4 km) $NO_2$ Column ($\times 10^{15}$ molec/cm$^2$)" for consistency.
  - Add a table summarizing key attributes of the datasets (e.g., resolution, uncertainties).
    - o Table 1 (shown above) summarizing the key attributes of all datasets has now been added to the manuscript.

**Section 3.1.1**
- The calculation of the 0–15 meter column concentration relies on surface NO2 measurements. Discuss the reliability of this assumption and its impact on comparisons. Quantify the average 0–15 meter column in absolute and relative terms.
  - o The mean 0–15 meter column for time periods coincident with the Pandora-DS data is $2.84 \times 10^{14}$ molec/cm$^2$. It represents 4.6% $\pm$ 3.1% of the modified Pandora-DS tropospheric $NO_2$ column (mean and standard deviation).
  - o We do not have information on how the $NO_2$ varies from 0 m to 15 m, and so it is difficult to discuss the reliability of this assumption and its impact on comparisons. This vertical resolution of 15 m is finer than the grids of the HeiPro and GEM-MACH datasets we are using. In the best-case scenario, the 0 m and 15 m $NO_2$ values are the same. In this scenario, we can use the NAPS in situ error estimate of 0.4 ppbv, converted to a column value from 0–15 m, to obtain an error estimate for the 0–15 meter column for each data point (mean error estimate of $1.53 \times 10^{13}$ molec/cm$^2$). On average, the errors represent 9.8% of the 0–15 meter column and 0.34% of the modified Pandora-DS tropospheric $NO_2$ column. In the worst-case scenario, the $NO_2$ value at 0 m drops to 0 ppbv at 15 m. In this extreme scenario, the mean 0–15 meter column for time periods coincident with the Pandora-DS data is $1.42 \times 10^{14}$ molec/cm$^2$. It represents 2.4% $\pm$ 1.7% of the modified Pandora-DS tropospheric $NO_2$ column (mean and standard deviation). On average, the error estimates (which comes from the 0.4 ppbv uncertainty in the in situ measurements) now represent 0.35% of the modified Pandora-DS tropospheric columns. Therefore, in the worst-case scenario, the 0–15 m column represents ~2% less of the modified Pandora-DS tropospheric columns currently used in the study. The effect that this has on Fig. 5, where the PBL contribution + missing 15 m contribution in the Pandora-DS measurements are quantified, is that the PBL contribution is effectively reduced by about half (e.g., from 19% mean relative bias to 8.5% at the SAA range of 66°–85°). Although not an insignificant reduction, this reduction will only occur with the unrealistic scenario that the surface $NO_2$ at some VMR drastically reduces to 0 ppbv at 15 m. Therefore, we now state in the manuscript the following on lines 554–557:

[revised manuscript text omitted]

---

## Author Comment (AC2)

**Response to AMT Reviewer #2**

We thank the reviewer for their helpful comments, which have helped to improve the manuscript. Our point-by-point responses are provided below in blue font, with new text added to the manuscript given in italics. Line numbers refer to the new clean version of the manuscript.

The paper "Retrieval of NO2 profiles from three years of Pandora MAX-DOAS measurements in Toronto, Canada" by Ramina Alwarda et al., presents investigations on the NO2 profiles retrievals from the offaxis measurement scheme of the Toronto Pandora instrument over 3 years (2018-2020). The retrieval is done using Optimal Estimation HeiPro profiling algorithm and comparing the obtained profiles and partial columns to official Pandora direct-sun measurements, in situ observations, satellite data, and an air quality forecasting model.

The authors find that the HeiPro surface NO2 are close to the in-stiu NAPS measurements (small under-estimation of less than 10%) and underestmating the GEM-MACH model surface value (up to 40%). The HeiPro partial columns (up to 4km) are larger than the satellite S5p data (in agreement to other validation studies), while they are much larger than the direct-sun tropopsheric estimate and from the GEM-MACH model.

The paper is interesting, in the scope of AMT, and I would suggest its publication after some revision.

In the current state of the manuscript, a comparison between different datasets is shown, but it is never clear if one of those datasets is considered as a reference, if it has already been validated elsewhere or if it is also prone to (large) uncertainties. The error are never mentionned, so it is not clear if the differences found between the datasets are within the combined uncertainties.

- The biases (both the multiplicative and mean relative) reported in the manuscript all include uncertainties based on the errors of each dataset. For example, the mean relative bias of HeiPro (0–4 km) partial columns to Pandora-DS tropospheric $NO_2$ is reported as 61% $\pm$ 9.7%. Therefore, even when considering the errors, there is a clear positive bias between HeiPro and Pandora-DS. Additionally, the Pandora-DS and TROPOMI tropospheric $NO_2$ data used in the study have been the subjects of previous air quality studies (e.g., Zhao et al., 2019; Zhao et al., 2022) and so we consider them as 'established' datasets. On the other hand, the HeiPro $NO_2$ dataset at this measurement site has not been previously studied or published. We have added Table 1 to the manuscript which outlines the key attributes of each dataset, including uncertainties. The table is copied below for reference.

**Table 1. Overview of the key attributes of the NO₂ datasets used in this study.**

| Dataset | Temporal resolution | Horizontal resolution | Errors/uncertainties |
|---|---|---|---|
| **Pandora-DS total column NO₂** | 90 s | < 4 km (for SZA < 50°) 4–17 km (for SZA 50°–80°) (Herman et al., 2009) | $1.3 \times 10^{15}$ molec/cm$^2$ (Herman et al., 2009, 2018) |
| **OMI stratospheric NO₂** | 1 day | 13 km × 24 km (Krotkov et al., 2017) | $2 \times 10^{14}$ molec/cm$^2$ (Krotkov et al., 2017) |
| **HeiPro** | 22 minutes | 5–10 km (effective pathlength, Ortega et al., 2015) | $4.4 \times 10^{14}$ molec/cm$^2$ (this work) |
| **TROPOMI tropospheric NO₂** | 1 day | 7 km × 3.5 km (5.5 km × 3.5 km since Aug. 2019) | $8.5 \times 10^{14}$ molec/cm$^2$ (Eskes and Eichmann, 2019) |
| **GEM-MACH** | 1 hour | 10 km × 10 km | N/A |
| **In situ NO₂** | 1 minute | Point measurement | 0.4 ppbv (Thermo Scientific, 2015) |

Another point is that a long investigation is performed to try to understand/quantify the causes of the differences between the Pandora MAX-DOAS and the Pandora direct-sun tropopsheric estimation, like investigating the influence of the PBL or the impact of different viewing angle due to the NO2 spatiotemporal heterogeneithy but the impact of some more basic assumptions are not really estimated and only very quickly discussed in the conclusion. In my opinion, two points are too briefly mentionned and not quantified enough:

- the choice of MAXDOAS long UV scans (oly discussed for the surface results)

- the quality of the Pandora direct-sun tropospheric estimation

We agree that the above two points are briefly mentioned. Please see below our more detailed explanations as to why we believe they are not significantly contributing to the observed biases.

- How would the comparisons for the partial columns be with the long VIS scans?
  - Please see below a plot for partial column comparisons between HeiPro long UV versus long vis as well as a table (now Table A1 in the Appendix) that summarizes the biases for HeiPro long vis and long UV versus partial columns from Pandora-DS, GEM-MACH, and TROPOMI. As shown in the figure and table, the results and biases are quite similar: (i) the scatter plot below shows a zero-intercept slope of $0.97 \pm 0.004$ and mean relative bias of $0.7\% \pm 5.9\%$, indicating minimal differences between the long UV and long vis partial columns, and (ii) the table below shows that the HeiPro long UV and long vis biases towards the partial columns are quite similar (mostly the same within uncertainties). We now state in the manuscript the following on lines 422–431:

*Although the direct-Sun and MAX-DOAS retrieval wavelength ranges are different due to the varying standard protocols for each, it is worthwhile to note that the HeiPro long vis versus long UV NO₂ partial column comparisons showed good agreement with one another, with a zero-intercept slope of 0.97 ± 0.004 and mean relative bias of 0.7% ± 5.9%. We therefore do not expect the choice of retrieval window to significantly impact the HeiPro long UV partial column comparisons to Pandora-DS (see Table A1 below for the HeiPro long vis partial column NO₂ comparisons).*

[Figure]

**Figure above (not in main manuscript or Appendix) shows the 0–4 km NO₂ column comparisons for HeiPro long vis vs. long UV.**

**Table A1. Multiplicative and mean relative biases (± uncertainties) of HeiPro towards NO₂ partial columns from Pandora-DS, TROPOMI, and GEM-MACH, for both the HeiPro long UV and long vis results.**

| NO₂ Partial Column Comparison | | HeiPro Scan Type | |
|---|---|---|---|
| | | long UV | long vis |
| **HeiPro vs. Pandora-DS partial columns** | Multiplicative Bias | 51% ± 0.8% | 49% ± 0.9% |
| | Mean Relative Bias | 61% ± 9.7% | 61% ± 6.8% |
| **HeiPro vs. TROPOMI** | Multiplicative Bias | 17% ± 4.0% | 13% ± 4.6% |
| | Mean Relative Bias | 37% ± 51% | 40% ± 45% |
| **HeiPro vs. GEM-MACH** | Multiplicative Bias | 12% ± 1.2% | 13% ± 1.3% |
| | Mean Relative Bias | 67% ± 7.1% | 64% ± 2.4% |

- How is the HeiPro MAX-DOAS comparing to the direct-sun total NO2 data?
  - We now include the figure below in the Appendix (Figure A4). Panel (a) shows the MAX-DOAS HeiPro comparisons to Pandora-DS total column NO₂. As expected, due to the larger

Pandora-DS columns now being compared, the bias is much lower (16% $\pm$ 0.69% multiplicative bias and 6.1% $\pm$ 4.8% mean relative bias). We now state the following in the manuscript on lines 461–464:

*For reference, HeiPro (0–4 km) vs. Pandora-DS total columns are compared in panel (a) of Fig. A4, which shows that HeiPro partial columns exhibit a positive multiplicative bias of 16% $\pm$ 0.7% and a mean relative bias of 6.1% $\pm$ 4.8%. Not surprisingly, there is better agreement here as compared to Fig. 3a (i.e., HeiPro vs. Pandora-DS tropospheric NO$_2$) since the Pandora-DS total columns are larger.*

[Figure]

**Figure A4**. Comparisons (2018–2020) of (a) HeiPro (0–4 km) NO$_2$ partial columns vs. Pandora-DS NO$_2$ total columns, (b) TROPOMI vs. Pandora-DS NO$_2$ total columns, and (c) TROPOMI vs. Pandora-DS tropospheric NO$_2$ columns. The dashed lines and color bar are as indicated in Fig. 3.

- how much is removed from the orginal direct-sun total NO2 dataset to create the tropopsheric dataset?

- The stratospheric NO$_2$ that is removed represents 34% $\pm$ 2.8% of the original Pandora direct-Sun total column NO$_2$ (this percentage is both the mean & median value). The following text has been added to the manuscript on lines 215–216:
  *The stratospheric portion that was removed accounted for 34% $\pm$ 2.8% of the Pandora-DS NO$_2$ total columns.*

- To get another estimate of the stratospheric-to-total column ratio at this measurement site, we use TROPOMI data (version 2.3.1) from 2018–2020 with a qa_value $\geq$ 0.75, within 10 km of the measurement site and 10 minutes of the Pandora-DS measurement time. The stratospheric-to-total column ratio is 44% $\pm$ 9.7%, which is within the uncertainty of 34% $\pm$ 2.8%, the amount of NO$_2$ removed using the Pratmo-OMI method. Please note that the stratospheric portion reported using TROPOMI is only during the satellite overpass time (13:30 LT), but we do not expect this value to change drastically if we were to incorporate data points across the entire day.

- how good is the OMI stratospheric estimation? and its diurnal evolution estimation from the model?

- The OMI stratospheric $NO_2$ data product we are using (version 3, Krotkov et al., 2017) has been used in previous publications (e.g., Choi et al., 2020, Zhao et al., 2022) and has been validated with satellite and ground-based instruments (Krotkov et al., 2017). Panel (c) of Figure A4 shows good agreement between TROPOMI and Pandora-DS tropospheric $NO_2$ (multiplicative bias: $-4.4\% \pm 3.5\%$; mean relative bias: $-0.9\% \pm 34\%$), indicating a reasonable stratospheric estimation at the measurement site in this study. The diurnal evolution estimation from the Pratmo model has been used previously in Zhao et al. (2019). More details of the performance of this method can be found in their Appendix B.

- how do the orginal direct-sun total NO2 compare to S5p total NO2?

- Figure A4 in the Appendix (shown above) now includes comparisons between Pandora-DS and TROPOMI total column $NO_2$ in panel (b) and tropospheric $NO_2$ in column (c). We now state in the manuscript the following on lines 461–475:

  *The TROPOMI vs. Pandora-DS $NO_2$ total and tropospheric column comparisons are shown in Fig. A4b–c, respectively. Pandora-DS and TROPOMI show good agreement with one another for both total column (multiplicative bias: $-12\% \pm 1.9\%$; mean relative bias: $0.1\% \pm 21\%$) and tropospheric $NO_2$ (multiplicative bias: $-4.4\% \pm 3.5\%$; mean relative bias: $-0.9\% \pm 34\%$). Note that the large uncertainties are due to the relatively large TROPOMI total column and tropospheric $NO_2$ errors. Additionally, the tropospheric $NO_2$ agreement in panel (c) provides more confidence in the stratospheric-tropospheric separation method that was used in the study (i.e., Pratmo-OMI data). TROPOMI total column $NO_2$ at this measurement site has been studied and validated in Zhao et al. (2020). Using the version 1 data product, Zhao et al. (2020) found that TROPOMI vs. Pandora-DS total column $NO_2$ had a zero-intercept slope of 0.70 and correlation coefficient of 0.75. The version 2.3 data product used in this work showed an improvement from version 1, with a zero-intercept slope of 0.89 and correlation coefficient of 0.81. The time period of the study in which version 1 was used (March 2018 to March 2019) was similar to that of this study (May 2018 to June 2020). Comparisons and validation of the newer version 2.3 TROPOMI data products are outside the scope of this work.*

These quantifications would allow to have more confidence in the Pandora retrievals and put the HeiPro data in relation to some reference data.

The summary in my understanding:

1) HeiPRO VCD (0-4km) are larger than: a) PGN-DStropo (by a lot!), b) S5p (by a quantity similar to what other validation in similar context has found), c) the GEM-MACH model. For the latter case, it is clear in Fig4 (although the larger differences are in winter months where maybe the number of points is not so representative?), but in Fig3c the comparison is more scattered and less clear.

- For the latter case (i.e., HeiPro vs. GEM-MACH), we believe that there is still a sufficient number of data points in winter months: $N_{spring} = 1827$; $N_{summer} = 2657$; $N_{fall} = 1574$; $N_{winter} = 646$.

The first 2 cases imply clear-sky conditions (direct-sun measurements for one, and some cloud filtering for the satellite pixels in the second case). Are the comparisons wrt model done with some kind of filtering too (ie cloud filtering)? is not, would this improve the comparison? Maybe you could try to see if the comparisons with the model improves if you only select the same comparisons pairs that are selected when comparing HeiPRO to the direct-sun (ie the clear sky cases)?

- Thank-you for this interesting suggestion. There was no cloud filtering done with the HeiPro vs. GEM-MACH comparisons. Although we do not have any cloud filtering/flags in the GEM-MACH data, we paired the GEM-MACH data with the HeiPro – Pandora-DS dataset, which would effectively ensure that GEM-MACH is now being compared to HeiPro under clear-sky conditions. The figure below shows the resulting comparison: the left panel is the original (as-is in the manuscript) while the right panel is the "clear-sky" case: there is only a small difference (panel (a) has a zero-intercept slope of 1.08 and mean relative bias of 60% $\pm$ 7.5%, panel (b) has a zero-intercept slope of 1.12 and mean relative bias of 67% $\pm$ 7.1%). Please note that because the HeiPro and Pandora-DS data are hourly averages, merging them with the GEM-MACH data does not fully ensure a clear-sky comparison since there may still be both clear and cloudy skies within one hour of measurements. In the manuscript, we have replotted Fig. 3a and 3c to show only the coincident data among HeiPro, Pandora-DS, and GEM-MACH, and so panel (b) below has replaced Fig. 3c.

[Figure]

**Comparisons between HeiPro (0–4 km) NO₂ partial columns (2018–2020) vs. (a) GEM-MACH (0–5 km) partial columns and (b) GEM-MACH (0–5 km) partial columns paired with Pandora-DS data. The zero-intercept linear regression (dashed blue line), ordinary least squares regression (dashed magenta line), and the 1:1 line (dashed green line) are depicted. The color bar indicates the normalized density of the data points. Figure not shown in main manuscript or Appendix.**

2) Heipro surface concentration are smaller than: a) NAPS (by 4.4%/9%), b) model (by 40% and 36%)

When the model is smoothed with the MAX-DOAS AVK, the agreement seems better. We only see it in Fig.8. Please give numbers for this case too, as it seems imoprtant to me to report on the differences when taking into account the vertical sensitivity of each technique.

- In the manuscript, we now state the following on lines 837–846:

  *Across all seasons, the HeiPro NO$_2$ median profiles from 0–200 m underestimate the unsmoothed GEM-MACH median values, while from 1.5–4 km, the HeiPro median profiles then overestimate the unsmoothed GEM-MACH median values. For the 0–200 m layer, the mean relative bias of HeiPro towards GEM-MACH decreases from −37% (unsmoothed) to −6.1% (smoothed). Note that these biases are representative of the integrated 0–200 m layer and may differ slightly from the surface values reported in Table 2. The most significant changes occur in the layer from 1.5–4 km, where the HeiPro bias towards GEM-MACH decreases from > 1000% (unsmoothed) to 2.6% (smoothed). The HeiPro surface underestimation and free tropospheric overestimation of the unsmoothed GEM-MACH profiles can probably be explained by the NO$_2$ inventories used in the GEM-MACH model, which, respectively, (i) utilize older inventories that do not account for reduced emissions over the years, and (ii) do not account for free tropospheric NO$_2$ sources while the a priori NO$_2$ profile contains free tropospheric NO$_2$.*

It would be good to add a table with the summary differences for all the cases.

- Thank-you for this suggestion. In the Conclusions section, we have added Table 2 (copied below for reference), which summarizes the biases for all the partial column and surface NO$_2$ intercomparisons presented in this work.

**Table 2. Summary of the multiplicative biases and mean relative biases (± uncertainties) for the HeiPro comparisons to partial columns and surface NO$_2$.**

| Datasets compared | Multiplicative Bias (%) | Mean Relative Bias (%) |
|---|---|---|
| HeiPro (0-4 km) vs. Pandora-DS tropospheric NO$_2$ | 51% ± 0.8% | 61% ± 9.7% |
| HeiPro (0-4 km) vs. TROPOMI tropospheric NO$_2$ | 17% ± 4.0% | 37% ± 51% |
| HeiPro (0-4 km) vs. GEM-MACH (0-5 km) NO$_2$ | 12% ± 1.2% | 67% ± 7.1% |
| HeiPro vs. NAPS in situ surface NO$_2$ | −5.8% ± 0.7% | −9.7% ± 7.5% |
| HeiPro vs. GEM-MACH surface NO$_2$ | −41% ± 0.5% | −37% ± 2.4% |
| HeiPro vs. GEM-MACH-smoothed surface NO$_2$ | −30% ± 0.8% | −6.0% ± 3.4% |

detailed comments:

[Figure]
* * *
- line 88-89: "Comparisons of these PGN sky algorithm data products with other datasets at the measurement site in this study will be the subject of a future study" --> this is a pity that is is not included here, it would have brought an interesting additional comparison.

- We agree that the inclusion of the PGN sky algorithm data products would have been an interesting additional comparison. However, current PGN sky data products are produced using the vis band fitting algorithm, not yet available for UV $NO_2$. We are working with PGN on a detailed validation and verification of UV $NO_2$ sky algorithm data products. But this is beyond the scope of current work (to validate the optimal-estimated-based retrieval).

- Fig1: I would move this figure when explaining the differences in pointing between the MAX-DOAS and the direct-sun. Also add in the caption what time-period has been averaged to create the S5p NO2 map.

- Figure 1 has now been moved to Section 2. The time period (2018–2020) was previously mentioned, but we have now specified which months (May 2018 – December 2020).

- line 194-> 197: the explanation of the stratospheric estimation substraction is presented here too quickly, and then again mentionned in Sect. 2.3.1. I would suggest to have one section explaining this part, with an illustration of the outcome and some quantification of the errors related to this step. When reading this part now, I have many questions coming up: why OMI and not S5p? and why that OMI product? is there any reference showing that it is a good stratospheric dataset? has it been validated?

- TROPOMI was not used to remove the stratospheric portion because we wanted to use an independent stratospheric data product that was not a part of the partial column intercomparisons. Additionally, according to Krotkov et al. (2017), the OMI stratospheric data product we used (v3) shows good agreement with the verification algorithm used for TROPOMI. Compared to v2 of the OMI stratospheric data product, v3 shows a reduced bias towards other satellite and ground-based datasets (Krotkov et al., 2017). Furthermore, this stratospheric $NO_2$ data product has been used in previous publications (e.g., Choi et al., 2020, Zhao et al., 2022) and has been validated with satellite and ground-based instruments. Lastly, as seen in Figure A4c, TROPOMI vs. Pandora-DS tropospheric $NO_2$ shows good agreement, providing further support for the Pratmo-OMI stratospheric-tropospheric separation method used in this study.

-For my own curiosity: are you performing zenith-sky twilight measurements with the pandora? if yes, you could also derive some stratospheric NO2 estimation from the Pandora itself...

- The Pandora does make zenith-sky twilight measurements. We will keep this suggestion in mind for future studies using stratospheric $NO_2$ at the measurement site.

- Pandora DS: I think it would be good to show its coherence wrt S5p total NO2 (eventually in the annex) - and discuss that this was done in the past, ie Zhao et al 2020, altough with a previous/different S5p product version.

- We now include Figure A4 in the Appendix, with Figure A4b showing the TROPOMI vs. Pandora-DS total column $NO_2$ comparisons. We also state the following on lines 469–475: *TROPOMI total column $NO_2$ at this measurement site has been studied and validated in Zhao et al. (2020). Using the version 1 data product, Zhao et al. (2020) found that TROPOMI vs. Pandora-DS total column $NO_2$ had a zero-intercept slope of 0.70 and*

*correlation coefficient of 0.75. The version 2.3 data product used in this work showed an improvement from version 1, with a zero-intercept slope of 0.89 and correlation coefficient of 0.81. The time period of the study in which version 1 was used (March 2018 to March 2019) was similar to that of this study (May 2018 to June 2020). Comparisons and validation of the newer version 2.3 TROPOMI data products are outside the scope of this work.*

- Sect. 2.2.2: explain a bit more the DOF from MAXDOAS, show an AVK and discuss more/show more results from both UV and VIS channel of the MAX-DOAS (not only decide because the long UV are in a closer agreement with the surface NAPS. By the way, I am not totally convinced by this statement. You base your analysis on the fact that the multiplicative factor in Fig A2a) is smaller than in A2b). But the regression slope and intercepts are smaller for A2b) thant for A2a)...

- To expand on the DOFS discussion, the following text has been added to the manuscript on lines 314–315:
  *…the degrees of freedom for signal (DOFS) for each profile, which represents the number of independent pieces of information obtained from the measurements…*

- We have additionally added the following text to the manuscript on lines 327–331:
  *Lastly, retrievals for which both the $NO_2$ profiles and aerosol extinction profiles had DOFS < 1 were excluded from the analysis for quality control purposes (e.g., Vlemmix et al., 2015) and represented 19% of the dataset. Such retrievals contain limited information from the measurements and are more influenced by the a priori profile. The mean DOFS values before and after filtering are $1.61 \pm 0.68$ and $1.88 \pm 0.42$ for aerosol extinction, respectively, and $2.35 \pm 0.49$ and $2.39 \pm 0.40$ for $NO_2$, respectively.*

- Regarding results from both MAX-DOAS UV and vis channels, we have added Table A1 to the Appendix, which summarizes the biases of HeiPro long UV versus $NO_2$ partial column comparisons as well as long vis versus $NO_2$ partial column comparisons.

- Regarding the averaging kernels, we have added Fig. A3 to the Appendix (copied below for reference), which displays averaging kernels for both $NO_2$ and aerosol extinction profile retrievals that meet the DOFS threshold, along with the respective dSCDs of $NO_2$ and $O_4$.

[Figure]

**Figure A3: MAX-DOAS dSCDs (UV, 338–370 nm) of (a) NO₂ and (c) O₄ retrieved on January 28, 2020. The right panels show the averaging kernels and respective DOFS for a single HeiPro profile retrieval of (b) NO₂ and (d) aerosol extinction. The dSCDs used in the NO2 and aerosol extinction profile retrievals are indicated by the red dashed lines in panels (a) and (c), respectively.**

- Regarding the regression slope and intercept for Figure A2a (long UV) vs. A2c (long vis), we identified that there was a small bug in the code whereby the long vis scans were being plotted as the short UV and vice versa. The corrected figure (Fig. A2) is now in the manuscript and is added below for reference. Long vis scans show larger slopes for both regressions and a larger intercept as well as a lower R value compared to the long UV. We have added the following text in the manuscript on lines 319–320:

  *Lastly, the long UV scans show improved regression slopes, intercept, and correlation coefficient to the NAPS in situ surface VMRs compared to the long vis scans (see Fig. A2).*

[Figure]

**Figure A2. Scatter plots for HeiPro surface NO₂ vs. NAPS in situ surface NO₂ VMRs for scan types and retrieval windows of (a) long UV, (b) short UV, (c) long vis, and (d) short vis. Only the long UV scans were incorporated in the results of this work. The dashed lines and color bar are as indicated in Fig. 3.**

Please also double check numbers. They are different in Fig A2a) and in in Fig 6a), while in my understanding, they are presenting the same datasets...

- The reason for the differences is because in Fig. 6a, the data points ($N$=6950) represent coincident data points among NAPS in situ, HeiPro, and GEM-MACH, while the data points in Fig. A2a ($N$=7322) represent coincident data points between NAPS in situ and HeiPro only. The number of data points in Fig. A2a has changed to $N$=7073 after fixing a small bug in the code.

- lines 223-215: the choice of only using the long scans make sense to me (more information content with larger number of elevation angles) and is confirmed by the larger DOFs and RMS (not clear what RMS is meant in line 228), but then the choice of using the long UV scans instead of the long VIS scans is less convincing to me. They are more in agreement with the surface NO2 (with the theory that the horizontal extent of the line-of-sight is smaller, ok), but this reason is not so applicable to the direct-sun Pandora geometry or the satellite and model extent.

- Thank-you for catching the use of "RMS" here. It is not RMS, but the HeiPro retrieved NO₂ partial column errors. We have fixed the axis in Figure A1 and any references in the text to indicate that these are partial column errors (lines 317 and 1018).
- The long UV scans were in closer agreement with the surface NO₂ due to the horizontal extent, while for the partial column comparisons, the biases of HeiPro long UV to the Pandora-DS, TROPOMI, and GEM-MACH partial columns were quite similar to those of HeiPro long vis to Pandora-DS, TROPOMI, and GEM-MACH partial columns. Because of the

similar performance to the partial columns, the agreement with the surface $NO_2$ became the deciding factor for which $NO_2$ retrieval window to use.

- I would show or at least comment the difference between results with both UV and VIS retrievals later on in the manuscript (at least as a form of error estimation?).

- As shown in the HeiPro long vis versus long UV scatter plot above and Table A1 immediately following it, there is a minimal difference between results with both long UV and long vis retrievals when comparing partial columns. For the most part, the biases relative to other datasets when using long UV and when using long vis are within the error estimates. We have now included the following in the manuscript (lines 422–431):

  *Although the direct-Sun and MAX-DOAS retrieval wavelengths are different due to the varying standard protocols for each, it is worthwhile to note that the HeiPro long vis versus long UV $NO_2$ partial column comparisons showed good agreement with one another, with a zero-intercept slope of 0.97 $\pm$ 0.004 and mean relative bias of 0.7% $\pm$ 5.9%. We therefore do not expect the choice of retrieval window to significantly impact the HeiPro long UV partial column comparisons to Pandora-DS (see Table A1 for the HeiPro long vis partial column $NO_2$ comparisons).*

- line 268: GEM-MACH partial columns 0-5km. If you have the profiles, why you calculate the partial column up to 5km and not 4 km, as the MAX-DOAS?

- The GEM-MACH standard dataset contained $NO_2$ profiles from 0–5 km and the integrated 0–5 km partial columns. While it makes more sense to only report 0–4 km GEM-MACH columns, this would require a re-integration of the partial columns. This can be done but was avoided because the GEM-MACH $NO_2$ VMRs from 4–5 km are $\ll$ 0.1 ppbv, with the mean VMR at this altitude range being 0.0188 ppbv. We did a test regarding how much the $NO_2$ from 4–5 km contributes to the GEM-MACH (0–5 km) $NO_2$ VCD by integrating the data from 4–5 km and dividing by the (0–5 km) $NO_2$ VCD. We found this value to be 0.293% (see histogram below). We therefore do not expect that the 0–4 km GEM-MACH columns will vary significantly from the 0–5 km GEM-MACH columns. We have added the following text to the manuscript on lines 672–674:

  *While the GEM-MACH standard dataset includes partial columns from 0–5 km, we did not generate a 0–4 km partial column because we found the GEM-MACH $NO_2$ VMRs from 4–5 km to be very small, and that the integrated $NO_2$ from 4–5 km constituted only 0.3% of the GEM-MACH (0–5 km) partial columns.*

[Figure]

**Histogram of the frequency of the percent contribution of the 4–5 km GEM-MACH partial column to the 0–5 km GEM-MACH partial column for NO₂ at Downsview. Figure not added to manuscript or Appendix.**

- line 269: are the emissions in the model differenet for 2018, 2019 and 2020? you say in the discussion/conclusion that the emissions are outdated, and probably too high...

- The anthropogenic emissions used before September 2018 were based on the 2010 Canadian Air Pollutant Emission Inventory (APEI), the 2011 US National Emissions Inventory (NEI), and the 1999 Mexican inventory; after September 2018, the emissions were updated using the 2013 Canadian APEI, the projected 2017 US NEI, and the 2008 Mexican inventory. Given that most emissions have been rapidly trending downwards in both Canada and the US, the statement that the emissions used in GEM-MACH are probably high is correct. For example, the newest GEM-MACH emission dataset, which is based on the projected 2020 Canadian APEI and the projected 2023 US NEI and was implemented operationally in November 2021, has 20% less Canadian $NO_x$ relative to the previous emissions (and 30% less $NO_x$ for the US).

- line 281: why you use York linear fit for fig 3a and b) and then switch to ordinary linear regression in fig 3c)? it is a bit confusing

- The ordinary linear regression was used in Fig. 3c because the GEM-MACH data does not have uncertainty estimates. Therefore, a York linear regression could not be used. This has now been stated in the manuscript on lines 413–416:
  *Three types of linear regressions are presented in Fig. 3: the first is the York linear fit (York et al., 2004) in which the uncertainties in both datasets are incorporated into the regression, the second is the zero-intercept linear regression, and the third is the ordinary least squares linear regression (used only for the HeiPro vs. GEM-MACH regression since error estimates were not available for GEM-MACH data).*

- fig3: here or in the annex, I would add a plot of HeiPRO vs PGN total direct-sun and of PGN total direct-sun vs S5p total NO2.

- Please see the figure above (added to the Appendix as Fig. A4) which plots (a) HeiPro (0–4 km) vs. Pandora-DS total column NO₂, (b) TROPOMI vs. Pandora-DS total column NO₂,

and (c) TROPOMI vs. Pandora-DS tropospheric NO₂. We have also added the following text to lines 464–469 to discuss and reference this figure:

*The TROPOMI vs. Pandora-DS NO₂ total and tropospheric column comparisons are shown in Fig. A4b–c, respectively. Pandora-DS and TROPOMI show good agreement with one another for both total column (multiplicative bias: −12% ± 1.9%; mean relative bias: 0.1% ± 21%) and tropospheric NO₂ (multiplicative bias: −4.4% ± 3.5% ; mean relative bias: −0.9% ± 34%). Note that the large uncertainties are due to the relatively large TROPOMI total column and tropospheric NO₂ errors. Additionally, the tropospheric NO₂ agreement in panel (c) provides more confidence in the stratospheric-tropospheric separation method that was used in the study (i.e., Pratmo-OMI data).*

- why not stop the GEM-MACH partial columns 0-5km to 4km for fig 3 c)? as said above, you could integrate the GEM-MACH profiles only up to 4 km, to have the same vertical extent than the MAX-DOAS.

- Please see the response above as this point was also mentioned earlier.

- line 399: " as well as the different horizontal sensitivities between the direct-Sun and multi-axis viewing geometries." --> you could refer here also to the HeiPRO data retrieved in the VIS long scans, which are sensitive to another (longer) horizontal path ?

- Although the HeiPro long vis data is indeed sensitive to a longer horizontal path, both the long UV and long vis data performed similarly compared to one another and compared to the Pandora-DS tropospheric columns (as shown in Table A1 above). The differing horizontal sensitivities between the UV and vis do not seem to be able to explain the results since they perform similarly.

- line 574: " this appears to skew the profile shape (see HeiPro profiles near 0 m in Fig. 8)," : I don't understand what your refer to.

- This has been reworded to the following (lines 816–819):
*When plotting the median profiles, as is done in Fig. 8, it appears as though the extrapolation to the surface does not have a sharper gradient towards the surface (see HeiPro profiles in Fig. 8), although this is only because the median values are plotted; aside from profiles with lofted layers, each individual profile has a larger increase from the 100 m layer to the surface.*

- line 590: "do not account for free tropospheric NO2 sources while HeiPro measurements have some sensitivity to such layers," --> if you show the MAX-DOAS AVK, you can quantify/refer to its sensitivity in those layers

- Although the MAX-DOAS NO₂ averaging kernels show some sensitivity to altitudes as high as 1.5 km at times, we cannot confidently state that this retrieval always has sensitivity to the lower free troposphere since the PBL is changing throughout the day. Therefore, we have changed this sentence to the following on lines 845–846:
*…do not account for free tropospheric NO₂ sources while the a priori NO₂ profile contains free tropospheric NO₂.*

- line 593: "it is worthwhile to note that discrepancies between HeiPro and GEM-MACH profiles can be explained by the model inventories, " --> "can PROBABLY be explained by the.." (you don't show it is the case, you only assume it, right?!)

- We have now added "probably" to all sentences mentioning the GEM-MACH inventory as a plausible explanation for the bias (line 843 and line 955).

- line 598: "the smoothed GEM-MACH profile shows better agreement with HeiPro as it accounts for the MAX-DOAS measurement limitations and vertical sensitivity." --> please give also estimations for this case.

- Please see the response above as this point was also mentioned earlier.

- line 700: "partial column RMS fitting residuals": what are those RMS? RMS between measured and modelles dDSCD in the OE inversion? please explain. DOFs are also never really explained, it is taken as granted that the reader know what they are...

- Thank-you for flagging this. As mentioned in an above point, what was plotted was the retrieved $NO_2$ partial column error. This has been corrected in Figure A1 as well as any referencing text.
- Regarding the DOFS, we provide some more information in the manuscript on lines 314–315:

[revised manuscript text omitted]

---

## Author Response (AR2)

**Author's Response to Editor**

Notification to the authors: Figures A5 and A6 require an explicit copyright statement for Google Maps. Please see https://publications.copernicus.org/for_authors/manuscript_preparation.html#mapsaerials

Thank-you for flagging this. Google Maps copyright statements have been added to the captions for Figs. A5 and A6. The captions are copied below:

**Figure A5. A map (© Google Maps) of the measurement site (Downsview, white circle) and surrounding Greater Toronto Area masked by the TROPOMI pixel-averaged tropospheric $NO_2$ field from 2018 – 2020 during (a) summer only and (b) winter only. The color bar indicates the $NO_2$ tropospheric column in molec $cm^{-2}$. The black line represents the Pandora instrument's multi-axis azimuth viewing angle (AVA) of 255° and the magenta line represents the mean direct-Sun viewing angle during the TROPOMI overpass time (13:30 LT). The average MAX-DOAS effective path length of 7.5 km is depicted by the black marker along the line of sight.**

**Figure A6. A map (© Google Maps) of the measurement site (Downsview, white circle) and surrounding Greater Toronto Area masked by the GEM-MACH averaged $NO_2$ field from 2018–2020 at (a) 7 a.m. LT, (b) 1 p.m. LT, and (c) 5 p.m. LT. The black line represents the Pandora instrument's multi-axis azimuth viewing angle (AVA) of 255° and the magenta line represents the mean Pandora direct-Sun viewing angle during the corresponding times. The average MAX-DOAS effective path length of 7.5 km is depicted by the black marker along the line of sight.**